# Constraining Representations Yields Models That Know What They Don't Know

**João Monteiro, Pau Rodríguez**\*, **Pierre-André Noël, Issam Laradji, David Vázquez**
ServiceNow Research
`{FirstName.LastName}@servicenow.com`
\*Currently at Apple.

## Abstract

A well-known failure mode of neural networks is that they may confidently return erroneous predictions. Such unsafe behaviour is particularly frequent when the use case slightly differs from the training context, and/or in the presence of an adversary. This work presents a novel direction to address these issues in a broad, general manner: imposing class-aware constraints on a model's internal activation patterns. Specifically, we assign to each class a unique, fixed, randomly-generated binary vector – hereafter called *class code* – and train the model so that its cross-depths activation patterns predict the appropriate class code according to the input sample's class. The resulting predictors are dubbed total activation classifiers (TAC), and TACs may either be trained from scratch, or used with negligible cost as a thin add-on on top of a frozen, pre-trained neural network. The distance between a TAC's activation pattern and the closest valid code acts as an additional confidence score, besides the default unTAC'ed prediction head's. In the add-on case, the original neural network's inference head is completely unaffected (so its accuracy remains the same) but we now have the option to use TAC's own confidence and prediction when determining which course of action to take in an hypothetical production workflow. In particular, we show that TAC strictly improves the value derived from models allowed to reject/defer. We provide further empirical evidence that TAC works well on multiple types of architectures and data modalities and that it is at least as good as state-of-the-art alternative confidence scores derived from existing models.

## 1 Introduction

Recent work has revealed interesting emerging properties for representations learned by neural networks (Papernot & McDaniel, 2018; Kalibhat et al., 2022; Bäuerle et al., 2022). In particular, simple class-dependent patterns were observed after training: *there are groups of representations that consistently activate more strongly depending on high-level features of inputs*. This behaviour can be useful to define predictors able to reject/defer test data that do not follow common patterns, provided that one can efficiently verify similarities between new data and common patterns. Well known limitations of this model class can then be addressed such as its lack of robustness to natural distribution shifts (Ben-David et al., 2006), or against small but carefully crafted perturbations to its inputs (Szegedy et al., 2013; Goodfellow et al., 2014).

These empirical evidences and potential use cases naturally lead to the question: *can we enforce simple class-dependent structure rather than hope it emerges?* In this work, we address this question and show that one can indeed constrain representations to follow simple, class-dependent, and efficiently verifiable patterns on learned representations. In particular, we turn the label set into a set of hard-coded class-specific binary *codes* and define models such that activations obtained from different layers match those patterns. In other words, class codes *constrain representations* and define a discrete set of valid internal configurations by indicating which groups of features should be strongly activated for a given class. At testing time, any measure of how well a model matches some of the valid patterns can be used to reject potentially erroneous predictions. Codes are chosen *a priori* and designed to approximate pair-wise orthogonality.

**Motivation.** The motivation for constraining internal representations to satisfy a simple class-dependent structure is two-fold:

1. Given data, we can measure how close to a valid pattern the activations of a model are, and finally use such a measure as a confidence score. That is, if the model is far from a valid activation pattern, then its prediction should be deemed unreliable. Moreover, we can make codes higher-dimensional than standard one-hot representations. Long enough codes enable us to represent classes with very distinct, hence discriminative, features.

2. Tying internal representations with the labels adds constraints to attackers. To illustrate the advantage of this scheme, consider that an adversary tries to fool a standard classifier: its only job is to make it so that any output unit fires up more strongly than the right one, and any internal configuration that satisfies that condition is valid. In our proposal, an attack is only valid if the entire set of activations matches the pattern of the wrong class, adding constraints to the attack problem and effectively making it harder for an attacker to succeed under a given compute/perturbation budget as compared to a standard classifier for which decisions are based solely on the output layer.

Intuitively, we seek to define model classes and learning algorithms such that *intermediate representations follow a class-dependent structure that can be efficiently verified*. Concretely, we introduce total activation classifiers (TAC): a component that can be added to any class of multi-layer classifiers. Given data and a set of *class codes*, TAC decides on an output class depending on which class code best matches an observed activation pattern. To obtain activation patterns, TAC slices and reduces (*e.g.*, sum or average) the activations of a stack of layers. Concatenating the results of the slice/reduce steps across the depth of the model yields a vector that we refer to as the *activation profile*. TAC learns by matching activation profiles to the underlying codes. At inference, TAC assigns the class with the closest code to the activation profile that a given test instance yields, and the corresponding distance behaves as a strong predictor of the prediction's quality so that, at testing time, one can decide to reject when activation profiles do not match valid codes to a threshold.

**Contributions.** Our contributions are summarized as follows:

1. We introduce a model class along with a learning procedure referred to as total activation classifiers, which satisfies the requirement of representations that follow class-dependent patterns. Resulting models require no access to out-of-distribution data during training, and offer inexpensive easy-to-obtain confidence scores **without** affecting prediction performance.

2. We propose simple and efficient strategies leveraging statistics of TAC's activations to spot low confidence, likely erroneous predictions. In particular, we empirically observed TAC to be effective in the rejection setting, strictly improving the *value* of rejecting classifiers.

3. We provide extra results and show that TAC's scores can be used to detect data from unseen classes, and that it can be used as a robust surrogate of the base classifier if it's kept hidden from attackers, while preserving its clean accuracy to a greater extent than alternative robust predictors.

## 2 TOTAL ACTIVATION CLASSIFIERS (TAC)

### 2.1 REPRESENTING LABELS AS CODES

We focus on the the setting of $K$-way classification. In this case, data instances correspond to pairs $x, y \sim \mathcal{X} \times \mathcal{Y}$, with $\mathcal{X} \subset \mathbb{R}^d$ and $\mathcal{Y} = \{1, 2, 3, ..., K\}$, $K \in \mathbb{N}$. Usually, model families targeting such a setting parameterize data-conditional categorical distributions over $\mathcal{Y}$. That is, a given model $f \in \mathcal{F} : \mathcal{X} \mapsto \Delta^{K-1}$ will project data onto the the probability simplex $\Delta^{K-1}$.

Alternatively, we will consider classes of predictors of the form $f' \in \mathcal{F}' : \mathcal{X} \mapsto [0, 1]^L$ with $L \gg K$, *i.e.*, models that map data onto the unit cube in $\mathbb{R}^L$. We thus associate each element in $\mathcal{Y}$ with a vertex of the cube. In other words, we represent class labels with a set of binary codes $\mathcal{C} = \{C_1, C_2, C_3, ..., C_K\}$, and training can be performed via searching $\arg\min_{f' \in \mathcal{F}'} \mathbb{E}_{(x,y) \sim (\mathcal{X}, \mathcal{Y})} D(f'(x), C_y)$, for a distance $D : \mathbb{R}^L \times \mathbb{R}^L \mapsto \mathbb{R}$. In doing so, we seek models able to project data such that results are close to the vertex corresponding to the correct class label in terms of a distance $D$. At prediction time, one can predict via $\arg\min_i D(f'(x), C_i)$ or leverage $D(f'(x), C \in \mathcal{C})$ to estimate confidence.

The main advantage of defining such a class of predictors and an encoding scheme for labels is the fact that *one can control the properties of the set $\mathcal{C}$ to maximize the discriminability of its elements*. This is not possible for more common cases such as models that project onto $\Delta^{K-1}$, where the set $C$ is given by one-hot codes. In fact, for a large enough code length $L$ and $D$ given by the $L_1$ distance, we can easily choose $\mathcal{C}$ such that $D(C_i, C_j) > L/2 \ \forall \ i \neq j$. That is to say that a model needs to make several mistakes (along different dimensions of the label code) for its predictions to flip.

## 2.2 MODEL DEFINITION

To realize the model class $\mathcal{F}'$ briefly introduced above, we will leverage the set of $n$-layered neural networks. The outputs of layer $i$ within any chosen $f' \in \mathcal{F}'$ are represented by $a_w^i$, $w \in [1, 2, 3, ..., W_i]$, where $W_i$ indicates the width of layer $i$ given by its number of output features. We then partition the set of representations $a_w^i$ into disjoint subsets (or *slices*) of uniform sizes, denoted by $S_l^i$, $l \in \{1, 2, ...\lfloor L/n \rfloor\}$. Intuitively, our goal is to make it so that groups of high-level features "fire up" more strongly depending on the underlying class of the input. To enforce that property, we consider the sequence of total activations of slices. That is, for the feature slice $S_l^i$ on layer $i$, we will consider its total activation

$$A_l^i(f', x) = \sum_{a_w^i \in S_l^i} \sum_{h', h''} a_{w, h', h''}^i, \tag{1}$$

where features are arbitrarily considered as 2-dimensional objects ($h'$ and $h''$ are, for example, spatial dimensions in convolutional architectures for images). Note that total activations are defined similarly for features of any dimension by simply reducing away the extra dimensions. The sequence of total activations obtained from slices across all layers, denoted $A(f', x)$, will be referred to as the activation profile of $x$. An illustration is provided in Figure 1, where a generic convolutional model has the outputs of its layers partitioned into slices $S_l^i$ which are then reduced to yield the sequence $A(f', x)$, the activation profile of $x$ as induced by $f'$. Figure 16 in the Appendix shows a Pytorch (Paszke et al., 2019) implementation of feature slicing and activation profile computation. Given a stack of layers yielding activation profiles of dimension $L$, a set of class codes of dimension $L$, and a distance $D$, we define total activation classifiers (TAC) such that they make the prediction

$$y' := \underset{i \in \mathcal{Y}}{\arg\min} \, D(A(f', x), C_i) \text{ with associated confidence } -D(A(f', x), C_{y'}). \tag{2}$$

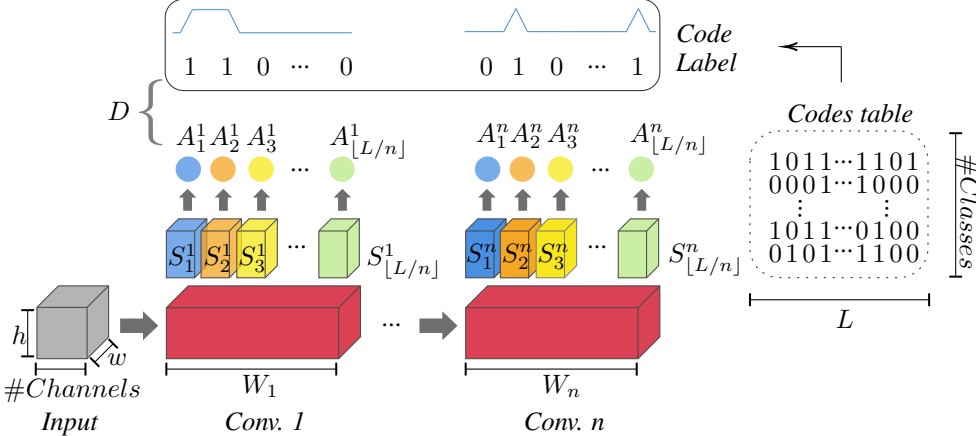

Figure 1: Illustration of a Total Activation Classifier (TAC). For simplicity, we illustrate the case where $a_w$ is 2-dimensional. The set of codes is chosen *a priori* and defines valid activation patterns.

## 2.3 TRAINING

Training is carried out to search for $f' \in \mathcal{F}'$ with activation profiles close to the correct code. To enforce that behaviour, we minimize the training objective

$$\mathcal{L}_{bin} = -\frac{1}{NL} \sum_{i=1}^{N} C_{y_i}^\top \log \sigma\big(A(f', x_i)\big) + (1 - C_{y_i})^\top \log \big(1 - \sigma\big(A(f', x_i)\big)\big), \tag{3}$$

where $N$ is the size of the batch of data used to compute the objective, $L$ is the length of the code and activation profile, and $\sigma$ is the sigmoid function. $\mathcal{L}_{bin}$ computes the binary cross-entropy loss for each dimension of the code and is minimized for a perfect match between activation profiles and the correct codes. While minimizers of $\mathcal{L}_{bin}$ yield the properties we need into a model, we empirically observed that training against $\mathcal{L}_{bin}$ is not trivial in that it requires the use of aggressive learning rates, rendering training unstable, especially so for large (*i.e.*, $\geq 1000$) label sets $\mathcal{Y}$. To overcome that issue, we define a second training objective by parameterizing a conditional categorical distribution over the label set using the distances between a given activation profile and all the codes and performing maximum likelihood estimation. More formally, the alternative training objective is

$$\mathcal{L}_{ce} = -\frac{1}{N} \sum_{i=1}^{N} \log \frac{e^{-D(A(f', x_i), C_{y_i})/\tau}}{\sum_{k=1}^{K} e^{-D(A(f', x_i), C_k)/\tau}}, \tag{4}$$

where $\tau$ is a scaling parameter.

While both $\mathcal{L}_{bin}$ and $\mathcal{L}_{ce}$ are reasonable choices for the training of TAC, each presents challenges. As mentioned above, $\mathcal{L}_{bin}$ is such that it yields unstable training but we found that, in the cases where one can minimize it, it produces models able to match codes very closely, and as such provide discriminative confidence scores. On the other hand, training $\mathcal{L}_{ce}$ is less unstable in that more common hyperparameter configurations work out of the box. However, minimizers of $\mathcal{L}_{ce}$ are not as good at matching vertices tightly since the objective only requires activation profiles to lie closer to the correct code than they are from other codes and, in that case, using distances as confidence scores is not as effective. We thus consider the linear combination of these training objectives

$$\mathcal{L} = \alpha \mathcal{L}_{bin} + \beta \mathcal{L}_{ce}, \tag{5}$$

where $\alpha$ and $\beta$ are hyperparameters. This choice takes advantage of the relative easiness of training against $\mathcal{L}_{ce}$ and the pressure of $\mathcal{L}_{bin}$ towards solutions able to match codes more closely. Remark that both objectives share minima, so minimizing their sum does not introduce any real trade-off.

## 2.4 Codes definition

The only requirement we consider for the class codes is that they are as dissimilar as possible in some sense. To build that set, one could use deterministic procedures such as, for instance, computing Hadamard matrices, but doing so adds constraints to the models since it implies codes of length $2^n, n \in \mathbb{N}$. We observed that, given large enough codes, randomly building $\mathcal{C}$ suffices for us to get discriminative codes. We thus define $\mathcal{C}$ as a random matrix of dimensions $|\mathcal{Y}| \times L$, where entries are independent $\text{Bernoulli}(0.5)$ random variables, observed at initialization. We remark that binary codes are chosen for simplicity. In particular, we can know exactly which features matter for each class at each layer, and this can be used for inspection of representations, or to determine features in the inputs implying predictions.

## 2.5 Attaching TAC to pre-trained models

Provided that one has their ready-to-use classifier, a TAC can be added on top of its features, and the base classifier is preserved as-is (*i.e.*, "frozen"). To do so, we isolate TAC from the base predictor via trainable projection layers applied in the representations of the base classifier. Then, we apply the slicing and reducing operations defined in Eq. 3 on those projections to obtain activation profiles. Projections correspond to independent MLP networks, one for each layer of the base classifier. The projection layers are then trained following the loss in Eq. 5, but gradients are not propagated to the base classifier, which remains unchanged. We remark that, while this approach is intended to simplify TAC training and enable its use in more practical scale, it also enables the combined use of confidence scores obtained by TAC and at the output layer of the base classifier (*e.g.*, the maximum softmax probability (MSP) (Hendrycks & Gimpel, 2016) or the maximum logit score (MLS) (Vaze et al., 2022)), *i.e.*, one can reject inputs that have too low confidence w.r.t. either of these easy-to-obtain scores. In other words, adding a TAC on top of an existing classifier can only improve upon any confidence score that was already in use.

## 3 Evaluation

Evaluations are split into three main parts:

**Section** 3.1: We start with a proof-of-concept and show that TAC can match activation patterns defined by class codes. We further show that small norm attackers are not able to match codes as well as clean data, rendering the distance between activation profiles and codes a good confidence score.

**Section** 3.2: We then proceed to the main evaluation and use TAC as an add-on to existing classifiers. In this case, we evaluate performance under the rejection setting and show TAC to improve upon the base classifier. We further evaluate TAC when used to detect test data from unseen classes.

**Section** 3.3: We seek additional applications of TAC and put it to test as a robust surrogate to the base classifier. We assume the threat model where only the base predictor is exposed to attackers, and observe that simple adversarial training results in competitive robust accuracy against strong attacks.

**Baselines**: For a fair comparison, we consider approaches that do not require access to auxiliary data of any sort. Directly relying upon statistics of the output layer of classifiers was observed recently to yield state-of-the-art performance on different OOD detection tasks (cf. MLS, (Vaze et al., 2022)). We thus consider MLS obtained from the base classifier as our main performance target, but also account for other similar recent methods for a broader comparison such as DOCTOR (Granese et al., 2021), as well as simple but effective baselines such as MSP (Hendrycks & Gimpel, 2016). In addition, we consider generative approaches that use likelihoods in the representation space as detection scores. For the robustness case, we use state-of-the-art robust classifiers as a reference of performance.

**Evaluation metrics**: Besides prediction accuracy, for the detection cases, evaluations are carried out in terms of the area under the operation curve (Det. AUROC) as well as in terms of the detection rate measured at the threshold where the false negative and false positive rates match, as commonly done to compute the Equal Error Rate. In Figure 15, we provide examples of the two metrics computation.

### 3.1 PROOF-OF-CONCEPT

To test for whether commonly used models are able to match activation patterns given by class codes, we train a TAC'ed WideResNet-28-10 (Madry et al., 2017) on CIFAR-10 (Krizhevsky et al., 2009), starting from random weights. TAC's slice/reduce operations are applied in three layers that output 160, 320, and 640 2-dimension feature maps. We use 16 slices in each layer so that the resulting code length $L$ is 48. Upon training to convergence, we inspect the representations obtained from a random draw of the test set, as displayed in Figure 2. Each column in the figure contains information about a different layer (layer depth grows from left to right). In the first row, we plot the raw activations after average pooling spatial dimensions. The activation profile $A$ is displayed in the second row, where one can see a tight match with the ground-truth codes, as displayed in the third row. The differences between $A$ and code are shown in the bottom row. We further test how good of a confidence score one can get by measuring some distance between $A$ and code. In Figure 3, we plot histograms of $L_1$ distances for clean data and adversarial perturbations of the test set of CIFAR-10. Attackers correspond to subtle PGD perturbations (Madry et al., 2017) obtained under a $L_\infty$ budget of $\frac{8}{255}$. We consider the white-box access model in which the attacker has full access to the target predictor and the table of codes. Attackers are created so that the activation profile $A$ moves towards the code of a wrong class. Attackers fail in matching codes as tightly as clean data. TAC can then spot attackers and defer low-confidence predictions. Confidence intervals for the distances at different layers are shown in Figure 10 in the appendix, showing similar behaviour throughout depth. We remark that we've empirically observed that one can sometimes improve performance by picking a particular layer, and a task/model dependent analysis could be carried out in a real application to select the target layer as reported for other cases in Figures 11a, 11b, 12a, and 12b.

### 3.2 ADDING TAC TO PRE-TRAINED CLASSIFIERS

**TAC preserves the accuracy of the base classifier.** We consider intent prediction tasks of DialoGLUE (Mehri et al., 2020). Namely, we conduct experiments on HWU64 (Liu et al., 2021), Banking77 (Casanueva et al., 2020), and CLINC150 (Larson et al., 2019), which correspond to sentence-level multi-class classification problems. We train base classifiers corresponding to the RoBERTa-BASE architecture Liu et al. (2019) to convergence. TAC is added afterward, and trained while the base model is frozen. We apply TAC across depth in 13 different points of the base model. The trainable parameters in this case correspond to the projection layers, one for each of the 13 layers, used to isolate the TAC operations from the pre-trained models. The number of slices in each layer is

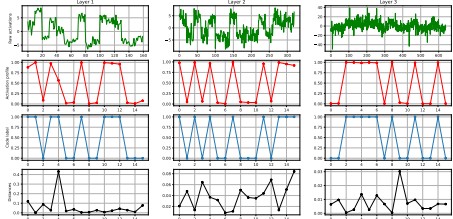

Figure 2: Activations of a TAC'ed Wide-ResNet-28-10 trained on CIFAR-10. Each column in the figure contains information about a different layer.

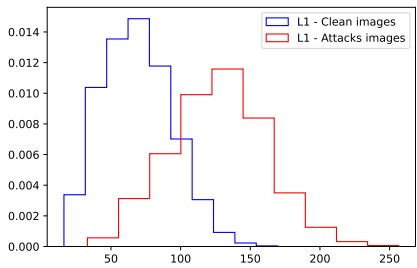

Figure 3: Distance histograms for attacks and natural images on CIFAR-10.

| | HWU64 | | | Banking77 | | | CLINC150 | |
|---|---|---|---|---|---|---|---|---|
| RoB-TAC | RoB-BASE | BERT++ | RoB-TAC | RoB-BASE | BERT++ | RoB-TAC | RoB-BASE | BERT++ |
| **93.9** | 92.6 | 92.9 | **94.2** | 93.9 | 93.4 | **97.4** | 97.0 | 97.1 |

Table 1: Prediction accuracy for three intent prediction tasks. TAC is added on pre-trained RoBERTa-BASE, and the base model is kept frozen during training. BERT++ indicates the best performance reported in Mehri et al. (2020) for each dataset.

treated as a hyperparameter and tuned with cross-validation. Projection layers correspond to stacks of fully-connected layers followed by ReLU activations, and features at each layer are extracted at the <eos> token. The depth of the projection stacks along with further details on the model architecture as well as on the datasets can be found in the Appendix, as well as additional results on image datasets further supporting our conclusions (cf. Table 6). Results are reported in Table 1. In this case, we compare the performance obtained from TAC's predictions with the base classifier it relies on (indicated as RoB-BASE in the table) as well as with the best performing case reported in Mehri et al. (2020) (BERT++). TAC outperforms the baselines in all three cases, which suggests the additional constraints imposed at early layers do not significantly affect representation capacity.

**Value analysis of predictors able to reject.** We evaluate TAC as well as different statistics of the output layer of base predictors when those are all treated as *rejecting* classifiers, *i.e.*, models that can abstain if not sufficiently confident in their predictions. We then leverage the framework proposed by Casati et al. (2021), which defines the application-specific *error cost* $\omega$ capturing the ratio of how much we dislike accepting incorrect classifications over how much we like accepting correct ones. This framework then proposes to evaluate different predictors in terms of their *value* $\mathcal{V} = \frac{N_c - \omega N_i}{N}$ as a function of $\omega$, where $N_c$ and $N_i$ correspond to the number of correct and incorrect error detection predictions made by a model over a sample of size $N = N_c + N_i + N_r$, for $N_r$ rejections. Note that the extreme case $\omega = 0$ has $\mathcal{V}$ maximized at $N_r = 0$, in which case this value matches the standard prediction accuracy. In Figure 4, we plot the Value Operating Characteristic (VOC, cf. definition in Casati et al. (2021)) curves for TAC, MLS, and MSP. We pre-trained a ViT Base-16x16 (Dosovitskiy et al., 2020) as the base predictor, and TAC operations are performed in 13 different layers throughout the model. We perform $k$-fold ($k = 5$) random splits on the validation set of ImageNet and, for a given split and value of $\omega$, we then use the $k - 1$ left-out splits to select the confidence rejection threshold that maximizes $\mathcal{V}$. Curves averaged over splits are plotted for the data used for threshold selection (indicated as *train* in the plot) as well as for the left-out splits. One can then note that TAC's scores yield the highest value $\mathcal{V}$ throughout a broad range of $\omega$, if not all of it.

While we defer most of the implementation details to the Appendix due to space constraints, we highlight a few practical observations. In particular, we observed that setting the distance $D$ to the $L_1$ distance along with using the full activation profiles and codes to yield a good default, working well across most cases. In some particular cases, on ImageNet specifically, we observed gains in moving to a cosine similarity and focusing on the last layer only. As these choices are case dependent, we recommend the default setting in general but adjusting to the specific data of interest should be done whenever feasible. All of our design decisions were made with left-out validation data.

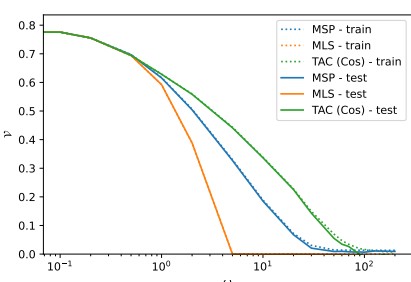

Figure 4: VOC curves for different predictors on ImageNet as a function of $\omega$, the application-specific *error cost* characterizing how undesirable it is to accept incorrect classifications. The overall value $\mathcal{V}$ of TAC-based predictors dominates for a broad range of $\omega$.

| Method | Det. AUROC (%) | Det. Rate (%) |
|---|---|---|
| **HWU64** | | |
| MPS | **89.39** | 81.43 |
| DOCTOR | **89.39** | 81.43 |
| MLS | 89.36 | 80.12 |
| TAC ($L_1$) | 89.25 | **83.25** |
| **CLINC150** | | |
| MPS | 90.56 | 82.77 |
| DOCTOR | 90.56 | 82.96 |
| MLS | 91.36 | 82.66 |
| TAC ($L_1$) | **93.25** | **85.70** |
| **ImageNet** | | |
| MPS | 80.74 | 73.63 |
| DOCTOR | 80.91 | 73.69 |
| MLS | 53.69 | 52.75 |
| TAC (Cosine) | **89.41** | **81.64** |

Table 2: Detection of prediction errors. Detection rates are measured at the threshold where false positive and false negative rates match.

To further assess how good different scores are when used to define rejecting classifiers, the performance of different scores for *detecting prediction errors* is reported in Table 2. TAC's distances as measured between activation profiles and codes yield a higher detection rate in all cases. Finally, accuracy vs. rejection level curves are shown in Figure 5 for different models trained on ImageNet (refer to Appendix A.7 for results on other datasets). The horizontal axis represents the fraction of the test set the model must predict from, and comparisons are carried out in terms of the area under such a curve, which would be 1 for a perfect model. We remark that predictions for the whole test set are fixed, and different results are given by how good different scores are in predicting errors.

**Out-of-distribution detection with TAC.** We further assess performance under the setting where test instances might belong to classes unseen during training. That is, for an unseen class, a safe model would be expected to abstain from classifying it. We test this behavior by using TAC's distances to decide when not to predict. Performance is evaluated on the CLINC150 benchmark. The training sample is such that an UNKNOWN label is used to indicate classes not represented in the remainder of the label set. We train our models on all classes except UNKNOWN, but try to detect unknown classes at testing time. Detection performance is reported in Table 3. We compare with MLS Vaze et al. (2022), recently shown state-of-the-art in this setting, and the approach introduced by Shao et al. (2020) where likelihoods of features under a Gaussian mixture model (GMM) are used as detection scores. TAC improves detection performance relative to those other unsupervised approaches that rely solely on the outputs of the base classifiers, especially so in terms of detection rate, indicating that one can leverage low-level outputs to design better confidence metrics. In Figure 8 in Appendix A.3, we report the detection performance obtained when different distances are used between TAC's activation activation profiles and class codes, and observe a somewhat consistent behaviour.

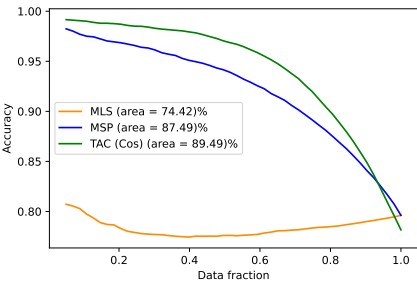

Figure 5: Accuracy of TAC'ed ViTs on ImageNet at different levels of allowed rejection.

| Method | Det. AUROC (%) | Det. Rate (%) |
|---|---|---|
| Shao et al. (2020) | 96.55 | 91.51 |
| Vaze et al. (2022) | 97.00 | 91.78 |
| TAC ($L_1$) | **97.28** | **92.82** |

Table 3: Comparison of different scoring strategies used for detection of examples from classes unseen during training. The detection rate measures the fraction of correctly detected attackers at the threshold where false positive and false negative rates match. For TAC, we report results considering the best performing distances, $L_1$ in this case.

## 3.3 ADDITIONAL APPLICATION: TAC AS A ROBUST SURROGATE TO THE BASE CLASSIFIER

TAC is meant as a way to detect prediction errors, **not** as a robust solution against *white-box* adversarial attacks (*i.e.*, for which the attacker has unlimited, direct access to the full model). In fact, our results (Table 4; more details below) indicate that TAC performs quite poorly in such a white-box adversarial context. That being said, we also consider a more limited adversary, *monitored white-box*, in which a black-box (*i.e.*, secret) TAC module monitors how a public, white-box model is being used. In other words, in the context of a monitored white-box API call to a publicly-available model, an outsider may observe that the API behaves identically to the white-box model, but the monitor may take actions following attacks, *e.g.*, request human assistance. Using TAC as a monitor proves to be a strong baseline, competing with the top performers for "true" white-box robustness.

Table 4 reports clean and robust prediction accuracy for Auto-Attack. Note that each of the two top rows report our results for both TAC and the base model, for both attack classes. The rest of the table provides reference results for the top-3 entries in Robust Bench's leaderboard (in grey in the table). Also note that auto-attack comprises a set of attack models evaluated individually. For completeness, for our models, we reported both the overall performance as well as results specific to each attack strategy. Adversarial training is employed using PGD perturbations obtained from the base classifier. To be clear, our goal here is not to outperform these robust baselines, and comparisons between the two access models are unfair in favor of the monitored white-box. Rather, we seek indications of other use cases where adding a TAC to an existing model might be useful. TAC preserves clean accuracy to a greater extent than the alternative approaches and, in the monitored white-box setting, TAC performs particularly well. In fact, up to improvements on the case of *apgd-dlr*, TAC is able to raise the performance of the base classifier to the level of the top performers in the leader board.

| Model | Attack class | Clean Accuracy (%) | Individual attacks from Auto-Attack (%) | | | | Average | Min. |
|---|---|---|---|---|---|---|---|---|
| | | | *apgd-ce* | *apgd-dlr* | *fab-t* | *square* | | |
| TAC / Base model | *Monitored w.-b.* | 93.23 / 95.59 | 90.76 / 0.01 | 53.74 / 0.00 | 75.86 / 0.00 | 74.34 / 3.48 | 73.68 / 0.87 | 53.74 / 0.00 |
| TAC / Base model | *White-box* | | 2.42 / 0.00 | 7.71 / 0.00 | 72.01 / 0.00 | 61.44 / 1.83 | 35.90 / 0.46 | 2.42 / 0.00 |
| Performance reference: Robust baselines | | | | | | | Overall Auto-Attack (%) | |
| Rebuffi et al. (2021) | *White-box* | 92.23 | – | – | – | – | 66.56 | |
| Gowal et al. (2021) | *White-box* | 88.74 | – | – | – | – | 66.10 | |
| Gowal et al. (2020) | *White-box* | 91.10 | – | – | – | – | 65.87 | |

Table 4: Robust accuracy of TAC under different access models for attackers. Depending on how much information the attackers has access to, an added TAC can raise the accuracy of the base model to close to state-of-the-art level. *Overall* for baselines should be compared with our *Min.* results.

## 4 RELATED WORK

**Error-correcting output codes.** ECOCs assign a unique code per class (Dietterich & Bakiri, 1994; García-Pedrajas & Fyfe, 2008; Rodríguez et al., 2018). Codes are commonly designed to maximize the Hamming distance between all pairs of classes so that errors can be detected and corrected by remapping incorrect predictions to the nearest code (Bautista et al., 2012). Verma & Swami (2019) and Song et al. (2021) leveraged ECOC to overcome adversarial attacks in multi-class classification. To do so, they train an independent binary classifier for each of the dimensions in a code. Rather than training an ensemble of binary classifiers, in our case we propose to slice the activations of already-existing architectures to obtain the elements of a code. As a result, TAC is significantly more efficient since it requires a single feature extractor. In addition, constraining outputs along the depth of a deep neural network makes adversarial perturbations more difficult to execute since it would require an attacker to simultaneously change the network's activations at multiple layers.

**Models that rely on low-level features.** The use of low-level (close to input) representations has been explored in recent work (Rodríguez et al., 2017; Evci et al., 2022; Monteiro et al., 2022a). However, their main goal is to improve the performance of classifiers rather than attaining more robust architectures. In Wu et al. (2021), on the other hand, classifiers are trained along with a decoder, and learned features are enforced to be such that inputs can be reconstructed. At testing time, reconstruction error can be used as a test statistic to detect anomalies. In Evci et al. (2022), features are collected throughout the model rather than at the outputs of a specific layer, which improves downstream performance under domain shift. Audio representations were shown in (Tang et al., 2019; Monteiro et al., 2022a) to improve downstream performance once low-level features are incorporated.

**Out-of-distribution detection.** OOD detection methods aim to detect samples that deviate considerably from the training distribution. Previous methods such as (Hendrycks & Gimpel, 2016; Liang et al., 2017; Hsu et al., 2020) focus on the maximum of the softmax activations, which tend to be smaller for OOD samples. OOD detection is also related to novelty detection (Abati et al., 2019; Perera et al., 2019; Tack et al., 2020) and anomaly detection (Hendrycks et al., 2018; Kwon et al., 2020; Bergman & Hoshen, 2020). Generative approaches were also considered in recent literature. Shao et al. (2020) and Jiang et al. (2022), for instance, train generative models on top of representations of in-distribution data and use the likelihoods of test examples as a confidence score. Most of these methods propose finding OOD samples in existing classifiers or propose new models to detect OOD examples explicitly. In contrast, we propose a new model component, allowing for better OOD detection by inspecting activations. The classification strategy discussed in (Monteiro et al., 2022b) also uses a slice/reduce type of representation. However, in their case, only a single layer is considered and activation patterns are matched to standard one-hot codes at the gradient level as opposed to the activations, so attackers still have room to find effective perturbations since, as with standard classifiers, any activation pattern apart from the output layer is valid, which we solve with TAC. More recently, Granese et al. (2021) and Vaze et al. (2022) showed that, in opposition to approaches that require access to auxiliary data (Jiang et al., 2018) or gradients (Liang et al., 2017), directly using simple statistics of the output layer of well tuned classifiers would outperform complex methods and reach state-of-the-art OOD detection performance on a number of benchmarks. Similarly to our proposal, Haroush et al. (2021) leverage class-dependent structure in representations across layers to detect out-of-distribution data. In that case however, pre-trained models are used and the focus lies on determining good detection scores. In our case, detection scores correspond to simple vector distances, and our focus lies on the training side. The two approaches are orthogonal and could be combined.

**Adversarial Robustness.** Adversarial attacks perturb the input in such a way that pushes the model's output towards an erroneous prediction. Several attack strategies were proposed specifically targeting the case of image classifiers (Goodfellow et al., 2014; Szegedy et al., 2013; Carlini & Wagner, 2017; Tu et al., 2019; Xiao et al., 2018; Zhang et al., 2020), but also for other domains, such as sentiment analysis systems (Jin et al., 2020), 3D point cloud models (Hamdi et al., 2020), audio recognition systems (Abdullah et al., 2019), and text classification (Morris et al., 2020; Zeng et al., 2020). Zhang et al. (2021) describe different strategies to make models more robust to adversarial attacks. Some examples are input transformation methods (Guo et al., 2017; Xie et al., 2018), stochastic defense methods (Cao & Gong, 2017; Papernot et al., 2016), and adversarial training (Goodfellow et al., 2014; Miyato et al., 2018). However, defense strategies cannot indicate that a system is under attack to carry out corrective/preventive measures. Thus, detection methods have been proposed such as supervised cases where a small binary classification network is used to that end (Metzen et al., 2017). Other supervised detection approaches operate by monitoring statistics of natural and perturbed samples (Feinman et al., 2017; Xu et al., 2017; Akhtar et al., 2018) or by combining outputs of indepdently trained classifiers and using that as inputs to a detector (Monteiro et al., 2019). Gradient-based detection methods were proposed such that adversarial data is spotted through the norm of the gradients of a model relative to the input (Lust & Condurache, 2020). For text classification, recent work (Rawat et al., 2021) generates out-of-domain samples to perform OOD detection.

## 5 CONCLUSION

We introduced TAC: a model component that can be attached to an existing predictor **without** affecting its accuracy to provide scores indicating how likely it is that predictions are incorrect. We showed that distances, as measured between activation profiles and class codes, define effective detection scores of problematic predictions. Notably, we successfully trained TAC on top of models as large as variations of BERT and ViT for text and image classification tasks respectively, and observed resulting scores to yield more effective rejecting classifiers than state-of-the-art alternatives that use the same base models TAC builds upon. We further showed that TAC can be applied to detect data from unseen classes and improves the detection rate relative to strong alternatives. We finally evaluated the robust accuracy of adversarially trained TAC on an additional task out of the original scope of our work to test for further applications where this kind of model can be useful, observing a promising performance when only the TAC component is kept private from attackers.

## REPRODUCIBILITY STATEMENT

In order to facilitate reproducibility of our empirical assessment, we developed all of our experiments on top of openly accessible data, models, and software tools. Moreover, both training and evaluation across all applications we considered were performed in single-GPU hardware. Implementation details are reported in Appendix B. In particular, code snippets of critical components are displayed in Figures 16 and 17.

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

## A  ADDITIONAL RESULTS

### A.1  FURTHER RESULTS ON TRAINING TAC FROM SCRATCH

In Table 5, we report clean prediction accuracy and detection performance for adversarial perturbations. In this case, we compare standard classifiers with their corresponding TAC considering both MNIST and CIFAR-10. PGD adversarial perturbations are obtained with Madry et al. (2017) attacks under $L_\infty$ budgets of 0.3 and $\frac{8}{255}$ for MNIST and CIFAR-10, respectively. The same WideResNet-28-10 discussed in Section 3.1 is used for CIFAR-10, while a 4-layered convolutional model is used for MNIST. We evaluated different detection scores corresponding $L_0$, $L_1$, $L_2$, and $L_\infty$ distances measured between $A$ and codes, and reported the one that performed the best in each case ($L_\infty$ and $L_1$ for MNIST and CIFAR-10, respectively). We evaluated both MSP and MLS for the base classifiers and reported the best performer. TAC improves the detection performance at the cost of a slight drop in prediction accuracy. We investigate the drop in accuracy by testing the *capacity* of the model after the TAC operations are included. To do that, we train models to overfit randomly assigned labels on the training set of MNIST. Figure 6 illustrates the in-sample error rate, where one can notice that while both models manage to achieve a minimum error rate, one class converges much faster than the other. Thus, we propose applying TAC to trained base classifiers to avoid such training difficulties. We'll show that, in that case, we avoid accuracy drops even in much larger models and datasets.

| | Clean Accuracy (%) | Det. AUROC (%) |
|---|---|---|
| **MNIST** | | |
| Base model | 99.0 | 75.7 |
| TAC ($L_\infty$) | 98.6 | 80.8 |
| **CIFAR-10** | | |
| Base model | 95.6 | 93.7 |
| TAC ($L_1$) | 94.9 | 95.7 |

Table 5: Prediction performance as well as adversarial detection evaluation for PGD attackers under $L_\infty$ budgets of 0.3 and $\frac{8}{255}$ for the cases of MNIST and CIFAR-10, respectively.

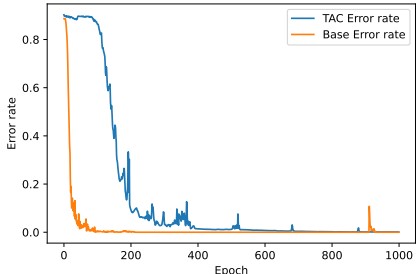

Figure 6: Capacity test on MNIST. Models overfit randomly assigned labels.

A larger size version of Figure 2 is displayed in Figure 7 for easier visualization.

### A.2  PREDICTION PERFORMANCE ON VISION DATASETS

In Table 6, we report prediction performance of TAC compared to the base classifiers it builds upon. Similarly to the cases of text classification reported in Table 1, the performance of the base classifier is matched by the additional component.

| MNIST | | CIFAR-10 | | ImageNet | |
|---|---|---|---|---|---|
| TAC | BASE | TAC | BASE | TAC | BASE |
| **99.3** | 99.0 | 95.0 | **95.6** | **80.6** | 80.0 |

Table 6: Prediction accuracy for three object classification tasks. TAC is added on pre-trained classifiers and closely matches their accuracy (referred to as BASE).

### A.3  COMPARING TAC UNDER VARYING $D$

In Figure 8, we report the detection performance obtained when different distances are used on top of TAC. Performance is consistent across different choices for $D$, except for the case of $L_\infty$, which indicates that there is always some dimension with a loose match between activation and code, no matter whether data is in- or out-of-distribution.

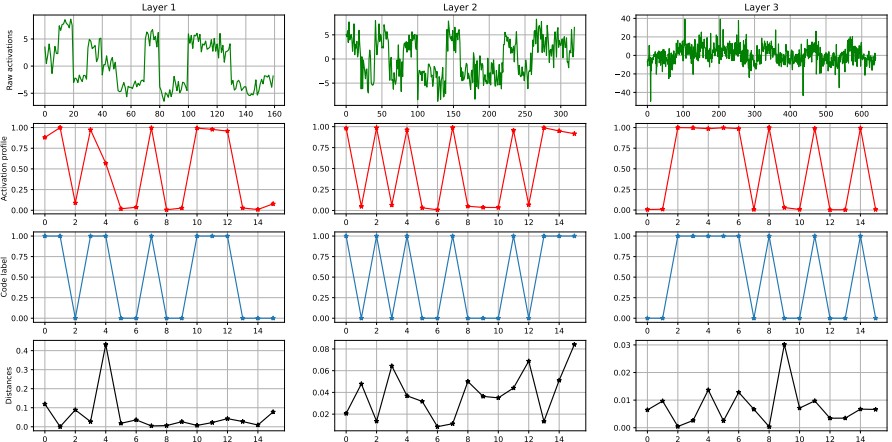

Figure 7: Activations of a TAC'ed Wide-ResNet-28-10 trained on CIFAR-10. Each column in the figure contains information about a different layer (layer depth grows from left to right). In the first row, we plot the raw activations after average pooling spatial dimensions. The activation profile $A$ is displayed in the second row, where one can see a tight match with the ground-truth codes, as displayed in the third row. The differences between $A$ and code are shown in the bottom row.

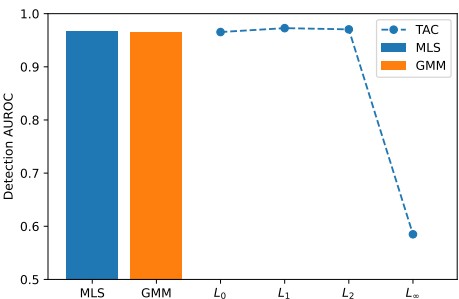

Figure 8: Detection of out-of-sample test instances in terms of AUROC.

### A.4 MATCHING OF ACTIVATION PROFILES AND CODES

In Figures 9a and 9b, we report further evidence showing that TAC succeeds in matching activation profiles and class codes. To do so, we compare codes with the set of class-wise average activation profiles given by the following for a particular class label $c \in \mathcal{Y}$ and a data sample $D$:

$$\bar{A}(f', c) = \frac{1}{|D_c|} \sum_{x,y \in D} A(f', x) \mathbb{1}[y = c], \tag{6}$$

where $\mathbb{1}[\cdot]$ is the indicator function and $D_c$ is the subset of $D$ that belongs to class $c$:

$$D_c = \{(x, y) \in D : y = c\}. \tag{7}$$

For both MNIST and CIFAR-10, we then build the heatmaps shown in Figures 9a and 9b by computing the cosine distances between class average activation profiles and codes. That is, a heatmap $H$ will be a $|\mathcal{Y}| \times |\mathcal{Y}|$ matrix such that entry $H_{ij}$ will be:

$$H_{ij}(f') = 1 - \cos(\bar{A}(f', i), C_j). \tag{8}$$

In both the cases of MNIST and CIFAR-10, we observe that $H$ is such that its main diagonal is highlighted, indicating an effective matching between activation profiles averaged for a given class and the corresponding class code.

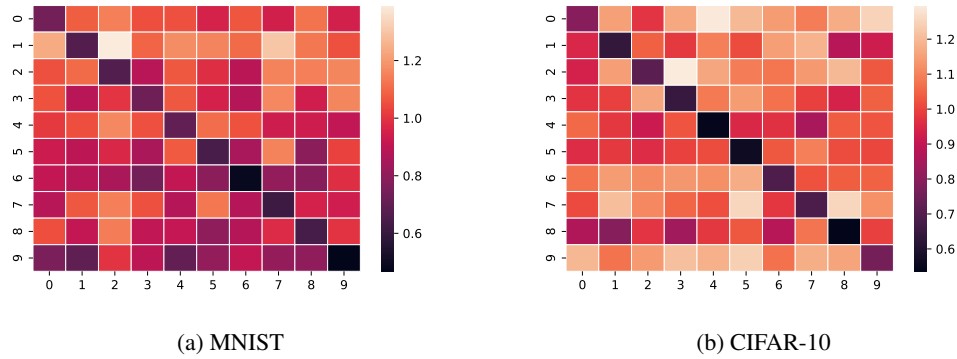

(a) MNIST                                    (b) CIFAR-10

Figure 9: Heatmaps indicating the distances between activation profiles averaged per class and different class codes.

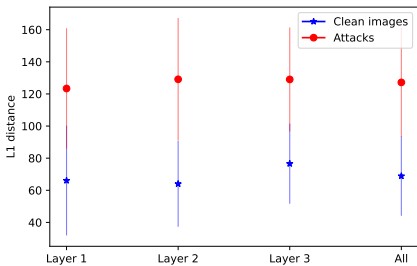

Figure 10: Layer-wise distance confidence intervals on CIFAR-10.

## A.5 PERFORMANCE ANALYSIS PER LAYER

In Figures 11a, 11b, 12a, and 12b we present per layer prediction accuracy along with a per layer error detection performance for the cases of CLINC150 and ImageNet. Interestingly, for ImageNet, we observe that performances grow monotonically with depth. That's not the case for CLINC150 where a non-monotonic relationship is observed and the peak is not at the last layer. These results support a general recommendation of setting as default the use of the full activation profiles and codes, but determine the best layers on data of interest whenever feasible.

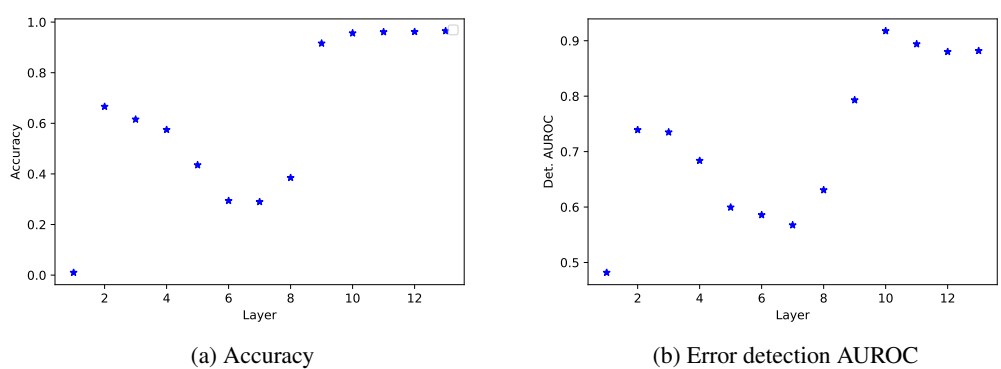

(a) Accuracy                              (b) Error detection AUROC

Figure 11: Performance at each layer individually for TAC trained on CLINC150.

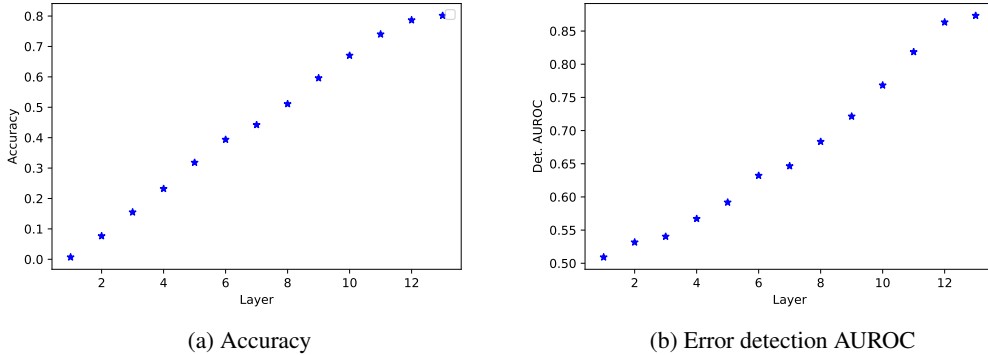

| (a) Accuracy | (b) Error detection AUROC |
|:---:|:---:|

Figure 12: Performance at each layer individually for TAC trained on ImageNet.

## A.6 ABLATIONS

Results on prediction accuracy and error detection performance are shown in the table below for ECOC baselines. We consider those to be ablations of TAC since they similarly project data to match binary codes, though they focus on the last layer only. In particular, we considered the extension of (Dietterich & Bakiri, 1994) proposed by (Rodríguez et al., 2018) where an ensemble of binary classifiers is trained on top of representations extracted at the last layer. The ensemble was trained on top of the same pre-trained base classifiers we considered for TAC, and the same hyperparameter sweep was accounted for. In addition, we further considered a variation of that baseline where the $\mathcal{L}_{CE}$ we proposed is further included in an attempt to obtain strong baselines.

Prediction accuracy was severely affected in all ablation cases, as was the error detection performance in two out of three datasets. Specifically in the case of ImageNet, we observed an improvement in error detection performance (at severe cost in prediction accuracy), which is inline with other experiments since we observed the last layer to yield the best performance only in this case.

| | Accuracy (%) | | Det. AUROC (%) | | Detection rate (%) | |
|:---:|:---:|:---:|:---:|:---:|:---:|:---:|
| Dataset | TAC | ECOC $/+\mathcal{L}_{CE}$ | TAC | ECOC $/+\mathcal{L}_{CE}$ | TAC | ECOC $/+\mathcal{L}_{CE}$ |
| HWU64 | 93.9 | 73.6 / 85.2 | 89.3 | 86 / 83.1 | 83.3 | 77.3 / 76.2 |
| CLINC150 | 97.4 | 83.4 / 87.8 | 93.3 | 83.5 / 82.6 | 85.7 | 75.6 / 75.1 |
| ImageNet | 80.6 | 74.4 / 77.8 | 89.4 | 90.06 / 83.0 | 81.6 | 83.5 / 74.9 |

Table 7: Ablation results. For ECOC, we present results with and without the use of $\mathcal{L}_{CE}$.

## A.7 EXTRA DETAILS AND RESULTS ON THE PERFORMANCE OF REJECTING CLASSIFIERS

### A.7.1 VOC CURVES

We provide further results comparing TAC with commonly used confidence scores such as output layer statistics when models are expected to reject or abstain from predicting if not confident enough. We then repeat the procedure we used to create the VOC curve reported in Figure 4 for other datasets. To do that, we split the test sets into 5 splits. For each such split, we use the larger portion of the data (four splits) to find the confidence threshold that maximizes the value $\mathcal{V}$ of the underlying predictor under the given error detection scoring strategy. We then plot the average value curves as a function of $\omega$ (the error cost) for both the larger (train) and smaller (test) data portions when we refrain from predicting from any exemplar for which the confidence score is below the threshold for the given $\omega$. Figures 13a, 13a, and 13c correspond to VOC curves for HWU64, CLINC150, and ImageNet (same as Figure 4 but with matching range of $\omega$ for consistency with other cases), respectively.

Results support our previous conclusion: the best detection score is application-dependent. In other words, best performers depend on the value of $\omega$ as well as on the underlying data. In any case, the use of TAC's scores results in improvements in several cases making it clear that the proposed approach

should be used in cases where prediction errors are costly/unsafe and rejecting is a possibility or a requirement.

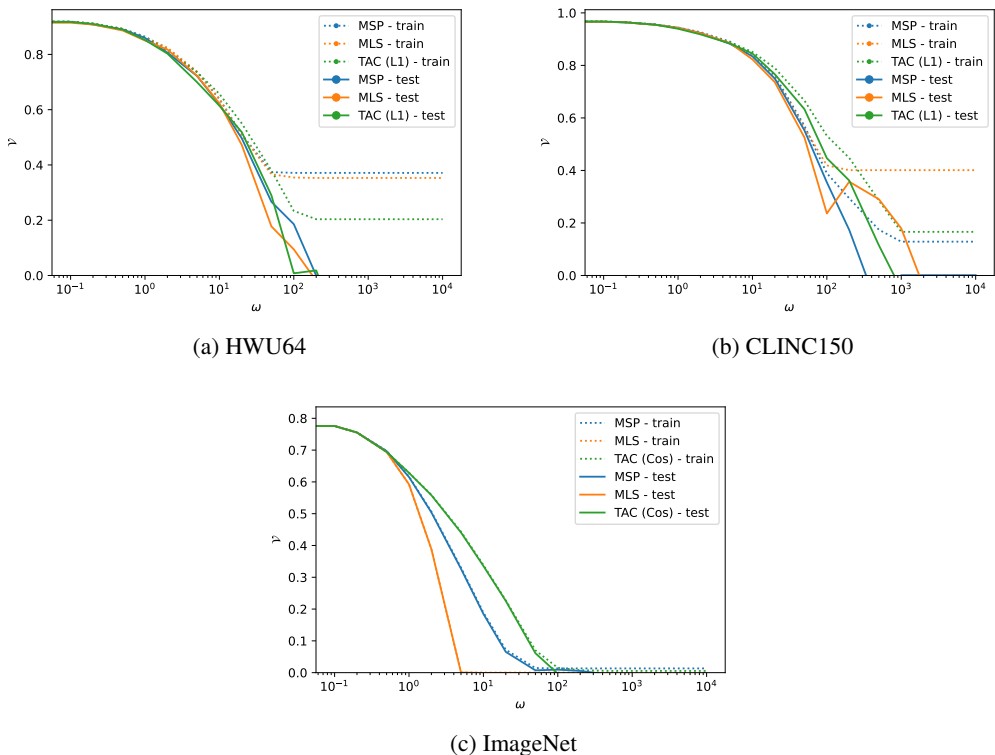

(a) HWU64              (b) CLINC150

(c) ImageNet

Figure 13: VOC Curves.

### A.7.2 Accuracy under varying rejection levels

To further assess the effect in performance given by rejecting classifiers when evaluated only on high confidence data points, we plot in Figures 14a, 14b, and 14c the test accuracy for various levels of rejection. In more detail, while in the vertical axis we plot the standard prediction accuracy on the underlying task, the horizontal axis corresponds to the fraction of the test set that is used for evaluation, and we keep the test instances yielding the highest confidence. For example, if *data fraction* is at the value $0.3$ in any of those curves, it indicates that $30\%$ of the test data was used for evaluation, and the selection is made after sorting data points in decreasing order of confidence. We further measure the area under the curves, which is better when higher and upper bounded by the value of $1$.

A general observation is that, for almost all cases, we observe an expected behaviour: the more we reject the better is the resulting prediction accuracy. This suggests that all of the considered confidence scores correlate well with prediction correctness. As before, results suggest that the best confidence score is data-dependent and also dependent on which level of rejection one is able to tolerate in a given application. Nonetheless, TAC can be beneficial over compared statistics.

### A.8 Adversarial detection on ImageNet

To further assess the utility of confidence scores derived from TAC, we verify their performance when used to detect decision-flipping adversarial perturbations in images from ImageNet Deng et al. (2009). We consider the black-box attack access model so that an independently trained classifier is used to generate attacks, and those are created from the validation partition of ImageNet. Three attack strategies are considered: FGSM Goodfellow et al. (2014) and PGD Madry et al. (2017), both under $L_1$ budgets, as well as very subtle $L_2$ CW Carlini & Wagner (2017) perturbations. Predictors

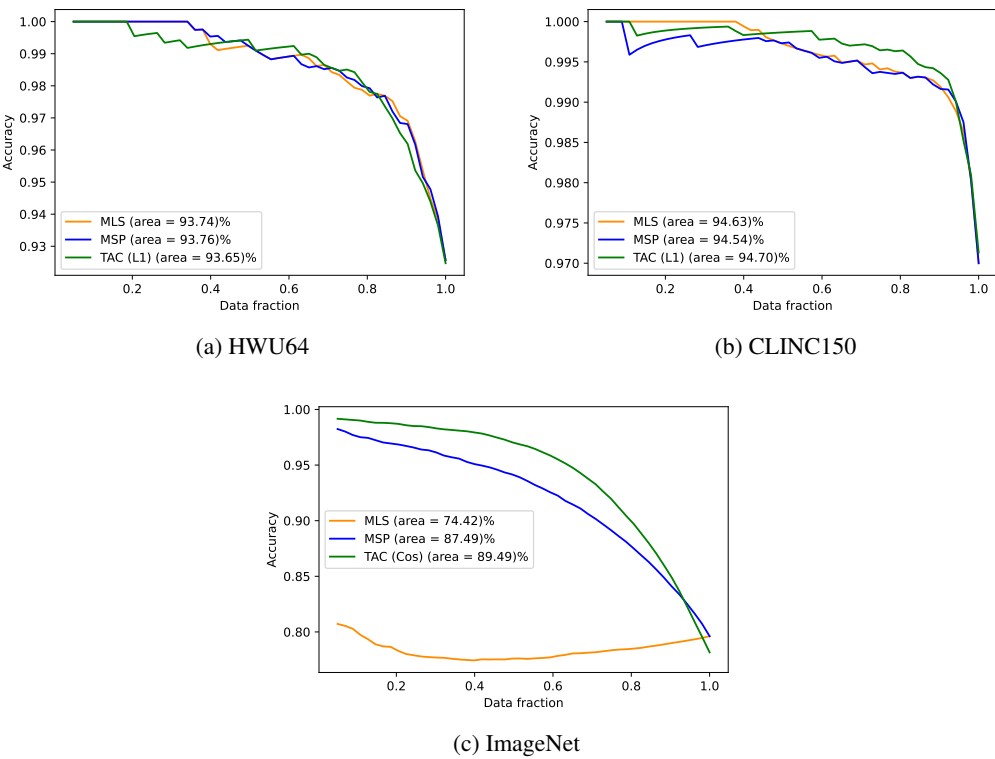

(a) HWU64          (b) CLINC150

(c) ImageNet

Figure 14: Accuracy/rejection curves. The horizontal axis indicates the fraction of the test set that was used to measure prediction accuracy. Only the highest confidence test points are used.

correspond to a pre-trained ResNet-50 where a TAC module is applied on top of projections of its features obtained at four different layers. Further experimental details are discussed in the Appendix. In Table 8, we compare the scores obtained from TAC with those obtained at the output layer of the base classifier. Namely, we compare $L_1$ distances measured between TAC's activation profiles and the code of the predicted class, with output layer statistics such as MPS and MLS. We note that $L_1$ distances are more discriminant of input perturbations than alternatives for the three perturbation cases we considered.

| | FGSM | | PGD | | CW | |
|---|---|---|---|---|---|---|
| | *Det. AUROC* (%) | *Det. Rate* (%) | *Det. AUROC* (%) | *Det. Rate* (%) | *Det. AUROC* (%) | *Det. Rate* (%) |
| MPS | 71.49 | 65.70 | 62.01 | 58.65 | 59.36 | 56.89 |
| MLS | 77.49 | 70.47 | 60.24 | 57.43 | 61.21 | 58.25 |
| TAC ($L_1$) | 77.94 | 70.90 | 63.11 | 59.67 | 61.92 | 58.92 |

Table 8: Performance on the detection of adversarial perturbations on ImageNet images.

# B   IMPLEMENTATION DETAILS

## B.1   ADDITIONAL TRAINING DETAILS

**Optimization and empirical observations**   Training was performed with Adam in all cases except for models trained on MNIST and CIFAR-10, where SGD with momentum was employed. A grid-search over hyperparameters was performed in each dataset, and models reported in the main text correspond to the configuration reaching the highest prediction accuracy on in-distribution validation data. All training runs were performed in single GPU hardware, which required a few hours for all cases except for models trained on ImageNet, which continued improving until 2-3 days. Overall, we noticed that TAC tends to perform better when weight decay is not applied or when its coefficient is

set to very small values ($< 10^{-5}$). Moreover, training against $\mathcal{L}_{bin}$ required relatively large learning rates compared to commonly used ranges for SGD. For MNIST and CIFAR-10 for instance, training to accuracy close to the base non-TAC model required learning rates greater than 1, which usually rendered training unstable. We observed this behaviour to change once we introduced $\mathcal{L}_{ce}$, after which common ranges of learning rate values used in popular recipes would work without change, though TAC still required moderate to low values of the weight decay parameter. Also, We usually needed to put pressure on one of the loss components, and we observed that the combination $(\alpha, \beta) = (1, 10)$ worked consistently well.

**Choices of distance functions**    To define $D$, we either use some $L_p$ distance for $p \in 0, 1, 2, \infty$ or a cosine distance. While we used $p = 1$ in most cases, we found that the choice of $D$ does not affect TAC's performance significantly. For ImageNet, we found that setting $D$ to the cosine similarity for this model resulted in the best-performing approach. Moreover, at testing time, we used only the deepest of the 13 layers to compute TAC's confidence scores since that resulted in improved performance.

**Mixup**    For models trained on CIFAR-10 and ImageNet, we performed Mixup (Zhang et al., 2017) between pairs of inputs and their corresponding codes. That is, given a pair of exemplars $(x', y'), (x'', y'')$, we compute the mixtures of data and codes as given by $x^{mix} = \alpha x' + (1 - \alpha)x''$, $C_{y^{mix}} = \alpha C_{y'} + (1 - \alpha)C_{y''}$, and $\alpha \sim \text{Beta}(0.2, 0.2)$. Pairs are created by pairing a training mini-batch with a random interpolation of its copy, and $\alpha$ is drawn independently for each pair. We present an implementation of the mixing approach we used for training in Figure 17.

**Adversarial perturbations**    Adversarial perturbations were created using Foolbox[1]. For experiments reported in Table 5, attackers correspond to PGD under $L_\infty$ distortions up to a maximum of 0.3 for MNIST and $\frac{8}{255}$ for CIFAR-10. Moreover, in this case, we assumed the white-box access model so that attackers have full access to our classifiers, including the code table. Attacks are then created so that activation profiles are moved towards the code of a wrong class (whichever one other than the correct). For the cases reported in Table 8 on the other hand, we consider the black-box case where an external predictor is used to generate attackers, which are then presented to our models. We created attacks using Torchvision's pre-trained ResNet-50[2]. In this case, attackers corresponded to $L_\infty$ FGSM and PGD, and $L_2$ CW. The perturbation budget given to attackers in each case was 0.05, 0.02, and 0.1 for FGSM, PGD, and CW respectively. Moreover, evaluation was performed on a subset of 10 images per class out of the validation sample of ImageNet.

Examples of the types of perturbations we consider are shown in Figure 18 where one can notice how subtle the considered transformations are, and hence how difficult of a problem it is to detect them. Except for the case of FGSM, one can barely notice any difference among those images by inspecting them visually, which is inline with the different detection performances we observed across attack strategies.

## B.2    MODEL ARCHITECTURES

### B.2.1    MNIST

Models trained on MNIST correspond to 4-layered convolutional stacks where each layer is followed by a LeakyReLU non-linearity. The numbers of channels in each layer are 64, 128, 256, and 512. To define TAC, we sliced post-activation features output by each such layer into 16 slices such that activation profiles and codes have dimension 64.

### B.2.2    CIFAR-10

Experiments on CIFAR-10 were carried out using a WideResNet-28-10 as described in Madry et al. (2017). Most ResNet's implementations (and its variants) split the model stack into four main blocks. We thus use the outputs of those blocks to define TAC. More specifically, in this case, we used the

---

[1]`https://foolbox.jonasrauber.de/`
[2]`https://pytorch.org/vision/stable/generated/torchvision.models.resnet50.html`

three blocks closer to the output, each outputting 160, 320, and 640 2-dimensional features. Once more, we defined TAC by slicing each set of features into 16 slices such that activation profiles and codes have dimension 48.

### B.2.3   IMAGENET

For ImageNet, we trained both a ResNet-50 as well as a ViT under the `BASE-16x16` configuration. For the ResNet case, we used the outputs of its four blocks to define TAC, and dimensions of representations output in each such part of the model are 256, 512, 1024, 2048, each split in 256 slices to compose activation profiles of size 1024. For the ViT, we used all the 13 transformer layers after averaging across the spatial dimension. That is, we collect 13 768-dimensional vectors across depth to define TAC, and set the number or slices in each layer to 256. As discussed in the main text, for the ViT case, we noticed that using a cosine distance during training helped improve performance. Moreover, at testing time, using only the final part of the activation profile and code resulted in the best scoring strategy in this case. In both models, base predictors are pre-trained and TAC is defined on top of projections of features as described in Section 3.2.

### B.2.4   TEXT CLASSIFICATION

Architectures for the three text classification datasets we considered corresponded to the `RoBERTa-BASE` configuration. We train base models and use the projection approach described in Section 3.2 to define TAC. As in the case of ViT, we collect 768-dimensional feature vectors across the 13 layers of the model. However, in this case, rather than averaging out the sequential component as in the ViT, we use only the elements corresponding to the end-of-sequence token, and 16 slices are considered in this case for each set of features.

### B.2.5   PROJECTION LAYERS

Projections used to enable the strategy described in Section 3.2 correspond to stacks of fully-connected layers followed by ReLU activations. Independent projections are defined in each layer used to feed TAC. We considered 5 projection configurations named `small`, `large`, `very-large`, `x-large`, and `2x-large`, and the choice amongst those options is treated as a hyperparameter to be selected with cross-validation for each dataset we trained on. The numbers of fully connected layers for each configuration is 1, 2, 3, 3[3], and 5. The `x-large` and `2x-large` configurations include `LayerNorm` operations in the inputs and outputs.

### B.3   TEXT CLASSIFICATION DATA

For text classification experiments, we train models on HWU64 Liu et al. (2021), Banking77 Casanueva et al. (2020), and CLINC150 Larson et al. (2019), which correspond to sentence-level multi-class classification problems. HWU64 is composed of 25716 sentences and contains 64 classes, and those correspond to different commands for virtual assistants. BANKING77 contains 13,083 data exemplars and 77 classes corresponding to different intents. CLINC150 contains 23,700 utterances and 150 different intent classes, being one of them an *out-of-scope* or UNKNOWN class, which is ignored during training of TAC, but used at testing time for detection evaluations.

### B.4   COST OF TRAINING

Training TAC on top of a pretrained model is rather fast. For instance, for the text classification applications, it takes only a couple of hours to train TAC on a single GPU. Here, much of the training time is actually the pretrained frozen model doing inference (so that its activations may be provided as input to TAC). If one were to cache these activations as a pre-processing step, this should provide a major speed-up on top of the already-quick training time.

---

[3]The `x-large` configuration includes `LayerNorm` operations in addition to the fully connected layers.

## B.5 DETECTION RATE IMPLEMENTATION EXAMPLE AND COMPARISON WITH THE EQUAL ERROR RATE

We present Python code snippets in Figure 15 showing that our choice of threshold used to compute the detection rate matches that of the standard Equal Error Rate.

```python
import numpy as np
from sklearn import metrics

def compute_detection_rate(
    y: list[float], y_score: list[float]
) -> float:

    """Computes the detection rate at the EER threshold.
    Args:
        y: Detection (binary) labels.
        y_score: Scores.

    Returns:
        float: Detection rate at a certain threshold.
    """

    fpr, tpr, _ = metrics.roc_curve(y, y_score, pos_label=1)
    fnr = 1 - tpr
    t = np.nanargmin(np.abs(fnr - fpr))
    return tpr[t]

def compute_eer(
    y: list[float], y_score: list[float]
) -> float:

    """Computes the Equal Error Rate metric.
    Args:
        y: Detection (binary) labels.
        y_score: Scores.

    Returns:
        float: Equal error rate.
    """

    fpr, tpr, _ = metrics.roc_curve(y, y_score, pos_label=1)
    fnr = 1 - tpr
    t = np.nanargmin(np.abs(fnr-fpr))
    eer_low, eer_high = min(fnr[t],fpr[t]), max(fnr[t],fpr[t])
    eer = (eer_low+eer_high)*0.5

    return eer
```

Figure 15: Python implementation of the detection rate we reported and the Equal Error Rate (EER). Our choice of threshold matches the threshold used for EER.

## B.6 TAC'S SLICE/REDUCE IMPLEMENTATION EXAMPLE

In Figure 16, we show an example of an implementation of TAC's slice and reduce operations on top of 2-dimensional features. These operations are repeated in all the layers that are to be TAC'ed in a model stack and results are concatenated to define the complete activation profiles, and those should match in dimension with class codes.

```python
def compute_activation_profile(
    features: torch.FloatTensor, n_slices: int
) -> torch.FloatTensor:

    """Compute TAC's slice-reduce operations.
    Expects features with shape [N,C,H,W] where:
        N is the batch size.
        C is the number of channels of the conv. layer.
        H and W are spatial dimensions.

    Args:
        features (torch.FloatTensor): Input features.
        n_slices (int): Number of slices.

    Returns:
        torch.FloatTensor: Sliced and reduced activations of a layer.
    """

    batch_size, feature_dimension = features.size(0), features.size(1)

    slices_size = feature_dimension // n_slices
    total_slices_length = slices_size * n_slices

    slices_indices = torch.arange(total_slices_length).view(n_slices,
        slices_size)

    activation_profile = features[:, slices_indices, :, :]
    activation_profile = activation_profile.view(batch_size, n_slices, -1)
    activation_profile = activation_profile.sum(-1, keepdim=True)

    return activation_profile
```

Figure 16: Pytorch implementation of feature slicing and activation profiles computation for a TAC induced by 2-dimensional convolution layers.

```python
def mixup_interpolation(
    data_batch: torch.FloatTensor,
    one_hot_labels: torch.FloatTensor,
    code_labels: torch.FloatTensor,
    interpolation_range: float,
) -> list[torch.FloatTensor]:
    """Mixup style data interpolation.

    Args:
        data_batch (torch.FloatTensor): batch of data.
        one_hot_labels (torch.FloatTensor): batch of labels in one-hot
            format.
        code_labels (torch.FloatTensor): batch of binary class codes.
        interpolation_range (float, optional): Concentration param for the
            Bet distribution.

    Returns:
        Tuple with batches of interpolated pairs from data_batch, codes,
            and labels.
    """

    permutation_idx = torch.randperm(data_batch.size()[0],
        device=data_batch.device)

    data_pairs = data_batch[permutation_idx, ...]
    label_pairs = one_hot_labels[permutation_idx, ...]
    code_label_pairs = code_labels[permutation_idx, ...]

    # Mixup interpolator: random convex combination of pairs
    interpolation_factors = (
        torch.distributions.beta.Beta(interpolation_range,
            interpolation_range)
        .rsample(sample_shape=(data_pairs.size(0),))
        .to(data_batch.device)
    )
    # Create extra dimensions in the interpolation_factors tensor
    interpolation_factors_data = interpolation_factors[
        (...,) + (None,) * (data_batch.ndim - 1)
    ]
    interpolation_factors_labels = interpolation_factors[
        (...,) + (None,) * (one_hot_labels.ndim - 1)
    ]
    interpolation_factors_code_labels = interpolation_factors[
        (...,) + (None,) * (code_labels.ndim - 1)
    ]

    # Interpolation for a pair x_0, x_1 and factor t is given by
    #     t*(x_0)+(1-t)*x_1
    interpolated_batch = (
        interpolation_factors_data * data_batch
        + (1.0 - interpolation_factors_data) * data_pairs
    )
    interpolated_labels = (
        interpolation_factors_labels * one_hot_labels
        + (1.0 - interpolation_factors_labels) * label_pairs
    )
    interpolated_code_labels = (
        interpolation_factors_code_labels * code_labels
        + (1.0 - interpolation_factors_code_labels) * code_label_pairs
    )

    return interpolated_batch, interpolated_labels,
        interpolated_code_labels
```

Figure 17: Pytorch implementation of Mixup interpolations.

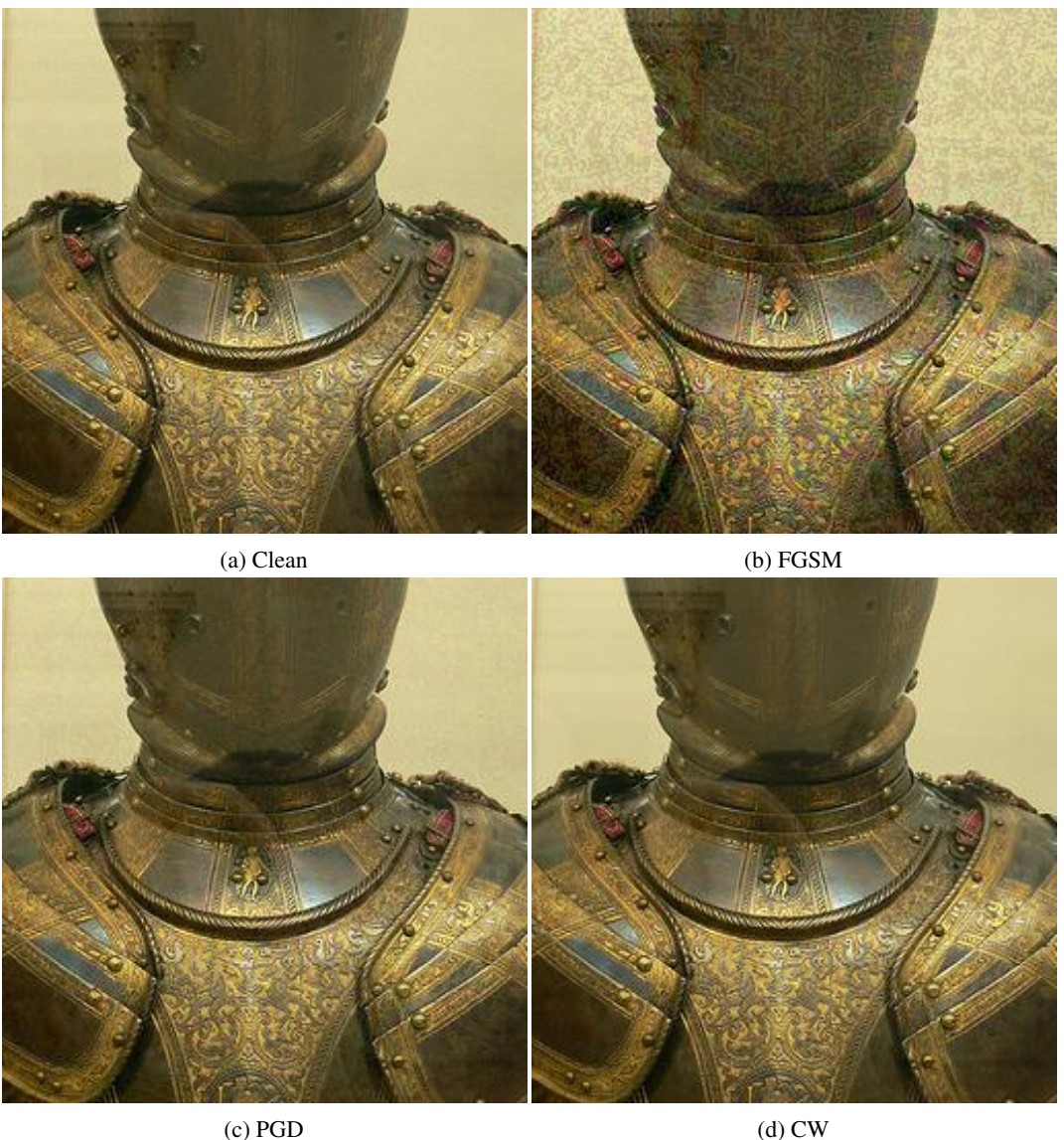

(a) Clean                             (b) FGSM

(c) PGD                              (d) CW

Figure 18: Examples of adversarial perturbations.

