# OpenReview forum: "Constraining Representations Yields Models That Know What They Don't Know"
_ICLR.cc/2023/Conference — ICLR 2023 poster_

### Official Review · Reviewer_rL3k · 2022-10-14

**Confidence:** 4
**Correctness:** 3
**Technical Novelty And Significance:** 2
**Empirical Novelty And Significance:** 3
**Recommendation:** 8

**Clarity, Quality, Novelty And Reproducibility:**

Clarity/quality - discussed above in the strengths and weaknesses.

Originality - as the authors point out, this is not too different from prior work on error-correcting output codes, but the specific way to chop up the activations seems new to me.

**Strength And Weaknesses:**

Thanks for the authors for the hard work on this paper.

Strengths
------------
- Appealingly simple idea
- Clear writing
- Strong performance
- Seems quite generally applicable

Weaknesses
----------------
- If I understand correctly, each of the binary codes is associated with a slice of the activations. However, activations in the earlier layers are generally worse at predicting the class (and perhaps one code in a vector of them). As such, shouldn't latter (in terms of neural network topology) codes be upweighted (in terms of loss computation) during training, during the distance computation (for OOD detection), and during inference? To assess whether the premises of my question is correct, it would be great to see a plot of (a) training error and (b) OOD distribution per layer: are earlier codes harder to learn than later ones and are they worse at OOD detection? You have something like this in Figure 2, but these are per-sample: how bad is the error vs depth overall? I think this statement in the appendix supports my concern: "Moreover, at testing time, we used only the deepest of the 13 layers to compute TAC’s confidence scores since that resulted in improved performance" - what fraction of the code is this? Did you have to do this kind of trimming for any other datasets? This is something important to understand and should be lifted out of the appendix.

- Sec 2.4 - but why *not* use Hadamard matrices? Is there any advantage in using random binary codes specifically?

- How easy is your method to adapt to multi-label classification tasks? How about regression? Would be nice to have a few sentences about this, but I'm not asking for any new experiments here.

- Table 2 - L1 is used for HWU64 and CLINIC150 but cos is used for ImageNet. It's noted in the appendix that you chose cos for ImageNet, Did you choose it on a held-out validation set? Is that the procedure you generally recommend? If so, please make explicit.

- The related work discusses error-correcting output codes. Why not compare to 1-2 of these methods that can be applied to neural networks? I understand they are not as efficient, but it's important to understand how good your performance is compared to previous similar ideas and trade that off with performance gains.

**Summary Of The Paper:**

The paper proposes the use of L-length binary codes (TAC) for the use of multi-class classification. Instead of having a one-hot representation, the authors' approach is to produce a binary value from each sliced-up section of the neural network's activations (Figure 1). Models can be trained to output TAC from scratch or via post-hoc add-ons to existing classifiers.

**Summary Of The Review:**

This is clear and interesting work. It needs to spend more time exploring how depth is related to code quality and utility for out-of-distribution detection, and needs to compare to other ECOC work. As is, the work is a 6. If both of my concerns are addressed, it would turn into an 8.

---

> ### Author Response · Authors · 2022-11-10
> **Author's responses 1/2**
>
> We thank the reviewer for the detailed assessment of our work. We are glad you find this contribution simple and generally applicable, which is our main goal. We tried our best to address the reviewer’s concerns in the following.
>
> > If I understand correctly, each of the binary codes is associated with a slice of the activations.
>
> Yes, that is correct.
>
> > activations in the earlier layers are generally worse at predicting the class (and perhaps one code in a vector of them).
>
> Yes, prediction accuracy grows with depth (not necessarily monotonically) since we train against an objective related to classification error. However, we are not explicitly training for identifying data variations (or anomalies for the lack of a better term), so different layers will be more or less discriminative of different types of variations in the data. If one knows exactly what kind of data variations are relevant downstream, then picking a specific layer might be helpful. If not, the default setting of considering the complete codes/activation profiles works well on a wide range of cases.
>
> In summary, in practice, we would recommend someone using a TAC to make that decision based on data they care about, but using the full activation profile/code works well enough in general.
>
> > To assess whether the premises of my question is correct, it would be great to see a plot of (a) training error and (b) OOD distribution per layer: are earlier codes harder to learn than later ones and are they worse at OOD detection?
>
> We would like to refer the reviewer to Figure 4 (now Figure 10), where we do this kind of layer-wise analysis for the case of CIFAR-10 under adversarial perturbations. The plot indicates 95% confidence intervals of detection scores computed independently for clean data and attacks, where one can see that the gap between clean data and attacks is significant no matter the layer.
>
> To provide further data, we present in the following tables the per layer prediction accuracy along with a per layer error detection for the cases of ImageNet and CLINC150. Interestingly, for ImageNet, we observe the behavior suggested by the reviewer and performances grow monotonically with depth. That’s not the case for CLINC150 where a non-monotonic relationship is observed and the peak is not at the last layer. We believe this observation supports a general recommendation of setting as default the use of the full activation profiles and codes, but determine the best layers on data of interest whenever feasible.
>
> Results are shown as more readable plots in the paper (see Figures 11 and 12).
>
> ImageNet:
>
> |    **Layer**   |   1   |   2   |   3   |   4   |   5   |   6   |   7   |   8   |   9   |   10  |   11  |   12  |   13  |
> |:--------------:|:-----:|:-----:|:-----:|:-----:|:-----:|:-----:|:-----:|:-----:|:-----:|:-----:|:-----:|:-----:|:-----:|
> |  **Accuracy**  |  0.7% |  7.6% | 15.5% | 23.2% | 31.8% | 39.4% | 44.2% | 51.1% | 59.6% | 67.0% | 74.0% | 78.7% | 80.1% |
> | **Det. AUROC** | 50.9% | 53.2% | 54.0% | 56.7% | 59.2% | 63.2% | 64.7% | 68.3% | 72.1% | 76.8% | 81.9% | 86.3% | 89.4% |
>
> CLINC150:
>
> |    **Layer**   |   1   |   2   |   3   |   4   |   5   |   6   |   7   |   8   |   9   |   10  |   11  |   12  |   13  |
> |:--------------:|:-----:|:-----:|:-----:|:-----:|:-----:|:-----:|:-----:|:-----:|:-----:|:-----:|:-----:|:-----:|:-----:|
> |  **Accuracy**  |  1.0% | 66.6% | 61.6% | 57.5% | 43.5% | 29.4% | 29.0% | 38.5% | 91.6% | 95.6% | 96.1% | 96.2% | 96.5% |
> | **Det. AUROC** | 51.8% | 73.9% | 73.5% | 68.4% | 59.9% | 58.6% | 56.7% | 63.1% | 79.3% | 91.8% | 89.4% | 88.0% | 88.2% |
>
>
> > I think this statement in the appendix supports my concern: "Moreover, at testing time, we used only the deepest of the 13 layers to compute TAC’s confidence scores since that resulted in improved performance"
>
> As shown in the tables above, the choice of relying on a particular layer or using the full activation profile seems application dependent, though using the full dimension seems to work well enough to be considered as a good default choice.
>
> > what fraction of the code is this? Did you have to do this kind of trimming for any other datasets? As suggested, this will be included in the main text rather than in the appendix.
>
> For all transformer-like architectures we considered, activation profiles obtained at each layer correspond to 1/13 of the full code length. This helped specifically in the case of ImageNet, though using the default setting still works well. We have mentioned this in the main text.

---

> > ### Author Response · Authors · 2022-11-10
> > **Author's responses 2/2**
> >
> > > Sec 2.4 - but why not use Hadamard matrices? Is there any advantage in using random binary codes specifically?
> >
> > We did not observe much of a gain in using Hadamard codes, and it introduces constraints that are inconvenient in some cases. Since our codes are long enough, random assignments yield (approximate) pairwise orthogonality, and we can use any desired dimension. Hadamard codes require lengths $L = 2^n, n \in \mathbb{N}$ which may require major architectural changes.
> >
> > > How easy is your method to adapt to multi-label classification tasks? How about regression? Would be nice to have a few sentences about this, but I'm not asking for any new experiments here.
> >
> > Regression is supported under discretization of targets. Multi-label classification seems a bit trickier and would require changes in the approach. One possible direction would be to have multiple code tables, and predict independently against each such table.
> >
> > > Table 2 - L1 is used for HWU64 and CLINIC150 but cos is used for ImageNet. It's noted in the appendix that you chose cos for ImageNet, Did you choose it on a held-out validation set? Is that the procedure you generally recommend? If so, please make explicit.
> >
> > We generally found that using L1 and the full code/activation profile are good default choices and work very well across all cases we considered. However, we observed gains in performance in some cases (in ImageNet specifically) when switching to a different distance or focusing on a particular layer.
> >
> > We did make design decisions using left-out data and would recommend doing so in practice, though the default setting should be good enough in most cases. We made sure to state that explicitly in the text.
> >
> > > compare to 1-2 of these methods that can be applied to neural networks?
> >
> > Thank you for the suggestion. We are currently working on experiments for the ECOC baselines/ablations and will include results in the paper and post them here once ready before the end of the discussion period. We decided to post the responses before those results are ready to start the discussion sooner rather than later.

---

> > > ### Author Response · Authors · 2022-11-14
> > > **Additional results**
> > >
> > > Results on prediction accuracy and error detection performance are shown in the table below for ECOC baselines. We consider those to be ablations of TAC since they similarly project data to match binary codes, though they focus on the last layer only. In particular, we considered the extension of [1] proposed in [2] where an ensemble of binary classifiers is trained on top of representations extracted at the last layer. The ensemble was trained on top of the same pre-trained base classifiers we considered for TAC, and the same hyperparameter sweep was accounted for. In addition, we further considered a variation of that baseline where the $\mathcal{L}_{CE}$ we proposed is further included in an attempt to obtain strong baselines.
> > >
> > > Prediction accuracy was severely affected in all ablation cases, as was the error detection performance in two out of three datasets. Specifically in the case of ImageNet, we observed an improvement in error detection performance (at severe cost in prediction accuracy), which is inline with other experiments since we observed the last layer to yield the best performance only in this case.
> > >
> > >
> > > |          | Accuracy (%) |                      | Det. AUROC (%) |                      | Detection rate (%) |                      |
> > > |----------|--------------|----------------------|----------------|----------------------|--------------------|----------------------|
> > > |          | TAC          | ECOC / ECOC+$L_{CE}$ | TAC            | ECOC / ECOC+$L_{CE}$ | TAC                | ECOC / ECOC+$L_{CE}$ |
> > > | HWU64    | 93.9         | 73.6 / 85.2          | 89.3           | 86 / 83.1            | 83.3               | 77.3 / 76.2          |
> > > | CLINC150 | 97.4         | 83.4 / 87.8          | 93.3           | 83.5 / 82.6          | 85.7               | 75.6 / 75.1          |
> > > | ImageNet | 80.6         | 74.4 / 77.8          | 89.4           | 90.06 / 83.0         | 81.6               | 83.5 / 74.9          |
> > >
> > > [1] Dietterich TG, Bakiri G. Solving multiclass learning problems via error-correcting output codes. Journal of artificial intelligence research. 1994.
> > >
> > > [2] Rodríguez P, Bautista MA, Gonzalez J, Escalera S. Beyond one-hot encoding: Lower dimensional target embedding. Image and Vision Computing. 2018.

---

> > > > ### Comment · Reviewer_rL3k · 2022-11-21
> > > > **Thank you**
> > > >
> > > > Thank you for the responses and additional experiments. I have raised my score.

---

### Official Review · Reviewer_eB98 · 2022-10-23

**Confidence:** 4
**Correctness:** 2
**Technical Novelty And Significance:** 2
**Empirical Novelty And Significance:** 3
**Recommendation:** 6

**Clarity, Quality, Novelty And Reproducibility:**

Although there are notable differences that make the proposed approach unique and interesting, there is more than a passing similarity to https://arxiv.org/pdf/2102.12967.pdf from ICLR 2022. This prior approach also involves extracting summary statistics from all layers of the neural network, and additionally has the benefit of proposing a principled statistical hypothesis testing framework for OOD detection.

**Strength And Weaknesses:**

Overall, this paper tackles an important problem and the experimental evaluations on robustness to adversarial attacks, ability to abstain, and out-of-distribution detection seem fairly convincing overall. However, I have a few concerns about motivation for the proposed approach as well as some lacking discussion (and perhaps experiments) of related prior work.

The approach is not well-motivated. In the introduction, the question is posed, “can we enforce this property rather than hope it emerges?”, but the reader is left wondering what property is actually under discussion. It would be helpful to include further discussion of the property of interest to motivate why it’s sufficient to target this to obtain “models that know that they don’t know.”

The use of binary codes is motivated by the need to have well-separated targets for optimization. However, in S2.4 it is stated that the codes are randomly selected with coordinates being independent binary random variables. Do the codes actually need to be binary?

**Summary Of The Paper:**

This paper considers the robustness of deep classification models, noting that out-of-the-box models can yield arbitrarily high-confidence predictions on out-of-distribution data. As a proposed remedy, “total activation classifiers” (TAC) are introduced to capture “class-dependent patterns” which are claimed to offer a reliable way to identify test data that do not conform to such patterns. Notably, the approach does not require access to out-of-distribution data at training time.

In more detail, TAC consists of a predictor mapping data onto the unit cube. The features are extracted from the neural network by taking “slices” from all intermediate activations which are then reduced (via summation) to 2-dimensional objects (e.g., spatial dimensions in convolutional architectures for images). The goal is to “make it so that groups of high-level features fire-up more strongly depending on the class of the input.” The sequence of all such slices is called the “activation profile” whose distance is measured to the classes. The class labels are represented as distinct binary codes; the set of binary codes is chosen in this way to maximise the “discriminability of its elements.” To train TAC, the binary-cross entropy loss is used to obtain class activation profiles which minimise the coordinate-wise distance to the corresponding class; variations on this loss are explored to improve training stability.

The overall TAC procedure can be applied to both “frozen” pre-trained models to obtain a secondary confidence score as well as trained “from scratch,” at a slight drop in accuracy (A.1). Thus, the recommended approach is to apply TAC to frozen models.

**Summary Of The Review:**

Overall, this paper tackles an important problem and the experimental evaluations on robustness to adversarial attacks, ability to abstain, and out-of-distribution detection seem fairly convincing overall. However, I have a few concerns about motivation for the proposed approach and there is an important omission in the discussion of related work.

---

> ### Author Response · Authors · 2022-11-10
> **Author's responses**
>
> Thank you for your comments and for pointing out relevant missing literature. We address the reviewer’s comments in the following:
>
> > Overall, this paper tackles an important problem and the experimental evaluations on robustness to adversarial attacks, ability to abstain, and out-of-distribution detection seem fairly convincing overall.
>
> We are glad that the reviewer found our evaluation convincing.
>
> > The approach is not well-motivated. In the introduction, the question is posed, “can we enforce this property rather than hope it emerges?”, but the reader is left wondering what property is actually under discussion.
>
> Thank you for pointing this out. We rephrased that part of Section 1 to make sure motivations are clearly defined. To clarify, the property we care about here is “imposing class-dependent structure in inner representations of neural networks”. More explicitly, we want models such that only a small number of internal configurations are observed, and such configurations are tied to classes in the label set.
>
> That property is useful since it adds constraints to attackers (standard unconstrained classifiers leave too much room for attackers), and it can be directly used to verify whether a prediction should be trusted (it shouldn’t be accepted if a valid pattern is not observed).
>
> > The use of binary codes is motivated by the need to have well-separated targets for optimization. However, in S2.4 it is stated that the codes are randomly selected with coordinates being independent binary random variables
>
> At a large enough code length, random codes are very close to being orthogonal with high probability. In fact, that’s true for any vector valued random variable of high enough dimension provided that each dimension has bounded variance. For reference, we point to a rather informal but precise discussion on this subject in the accepted answer here: https://math.stackexchange.com/questions/995623/why-are-randomly-drawn-vectors-nearly-perpendicular-in-high-dimensions.
>
>
> We did experiment with deterministic procedures that would guarantee pairwise orthogonality (e.g., taking rows of a Hadamard matrix), but that doesn’t improve upon random codes for the dimensions we consider and imposes constraints over the code length (Hadamard matrices require codes of length $2^n, n \in \mathbb{N}$), which can be inconvenient in practice.
>
> > Do the codes actually need to be binary?
>
> No. However, binary codes are convenient in practice since they are simpler. In particular, we can know exactly which features matter for each class at each layer, and this can be used for inspection of representations, or to determine features in the inputs implying predictions.
>
> This also makes our proposed approach very flexible in that the only requirement in our setting is for codes to be pairwise orthogonal, and we could obtain those codes from a continuous set by simply sampling from an uniform distribution over such a set (given a large enough dimension). We now mention that in the text.
>
> > similarity to https://arxiv.org/pdf/2102.12967.pdf
>
> We thank the reviewer for pointing out this relevant work. We have included it in the references. There are significant differences between our work and the suggested reference, and we see the two approaches as complementary rather than competing (**we focus on training** improvements not requiring a validation set; **they focus on testing** improvements requiring a validation set):
>
> **Methodological differences**: the approach in the referred paper doesn’t impose structure into representations, which is our main contribution. They leverage the fact that some channel/layer-wise structure emerges. I.e., they rely on pre-trained convolutional models and design detection statistics. In our case, we designed a training strategy to build that property into the architecture, while detection statistics are simply defined via vector distances. The two approaches are then complementary/orthogonal. Their results however further motivate the model classes we propose, and their approach could benefit from models we discuss in the paper as much as we could improve performance by introducing their statistics.
>
> **Efficiency differences**: TAC requires only the code table in addition to the model in order to compute detection statistics. Their proposal, on the other hand, requires one to score test instances against a validation set that needs to accompany the classifier downstream.

---

### Official Review · Reviewer_sw7u · 2022-10-25

**Confidence:** 4
**Correctness:** 2
**Technical Novelty And Significance:** 2
**Empirical Novelty And Significance:** 3
**Recommendation:** 6

**Clarity, Quality, Novelty And Reproducibility:**

The main themes and methods of the paper are very clear. However, the presentation of experimental results can be improved. For instance, metrics like baseline model accuracy and TAC model accuracy are reported for the intent classification tasks, but not for the vision tasks (e.g., CIFAR-10). Or if they are presented, they are presented in a different format such as the plots of figures 6 and 11.

In the mis-prediction detection experiments, the "Det. AUCROC" metric does not seem to be described, nor does the rationale for reporting the detection rate at the point where the "true positive rate (TPR) equals the false negative rate (FNR)". Why not use the point at which the TPR and FPR are equal?

Similarly, the results for the RobustBench leaderboard in table 4 are not well described and hard to make sense of. Why are all the cells for the baseline methods left blank? And what does the multicolumn column header "Individual attacks from from Auto-Attack (%)" mean?


**Strength And Weaknesses:**

Strengths:
- TAC is a simple extension that allows one to add safety and robustness to new and existing neural network models.
- Experiments with TAC show promise in terms of classification accuracy and mis-prediction detection across a range of datasets and misprediction penalty ranges.
- In gray-box adversarial settings, experiments show that TAC is effective at detecting adversarial examples.

Weaknesses:
- TAC is an empirical method with intuition, but not theory, justifying the use of ECOCs output by multiple network layers for preventing against spurious predictions. The paper's experiments are suggestive, but are not extensive enough to determine whether TAC provides robust and superior decision support across a wide range of tasks, and compared to other commonly used methods like temperature calibration.
- While the value-of-classification curve (VOC) was cool and something I have not seen before, many of the experimental results regarding misclassification detection rates are presented in a manner that is rather opaque and hard to interpret in real terms (see comments on paper clarity below).

**Summary Of The Paper:**


This paper presents a robust multiclass classification "add on" scheme for deep neural networks called TAC (total activation classifiers) in which multiple hidden layers of a network are trained to output error correcting codes (ECOCs) that are used to identify the proper class of an example and guard against spurious predictions in the case of out-of-distribution (OOD) examples and adversarial examples designed to trick the underlying model. Because ECOCs require more bits to be correctly predicted when determining the class label, TAC-augmented models are claimed to be more robust. Specifically, with TAC, the hamming (or cosine) distance to the closest code in the code book can used as a measure of model confidence, and predictions with at least a certain distance can be flagged for review or rejected.

Toy experiments on CIFAR-10 show that TAC-augmented neural networks can be trained to output ECOCs at multiple network layers, and that the distances of predicted codes is discriminative at identifying adversarial examples (though not perfectly so).

Language model experiments on 3 intent-classification tasks show that TAC improves accuracy over and above the original pretrained RoBERTA and BERT++ models. And experiments on ImageNet data with a pretrained ViT model shows that TAC better identifies missed predictions than other confidence detection measures like the max-logit-score (MLS), maximum-softmax-prediction (MPS), and a recently published approach called "DOCTOR".

Finally, on an adversarial benchmark dataset (RobustBench), TAC performs well at identifying attacks in a gray-box scenario where the attackers know the inner workings of the base neural network, but do not have access to the classification codebook. In the full white-box scenario where attackers have the code book as well, TAC is not competitive in providing protection against adversarial attacks.


**Summary Of The Review:**

Overall, this paper presents a simple and promising extension for adding robustness to both new and existing deep learning models. Theoretical justification is lacking, and some of the experimental results are hard to interpret in real terms. At the same time, some of the experimental results are compelling and show real improvements above commonly used SOTA baselines for misprediction and out-of-distribution identification. Therefore, I am in favor of accepting this paper to ICLR, but not strongly so.

---

> ### Author Response · Authors · 2022-11-10
> **Author's responses 1/2**
>
> We thank the reviewer for their time and for providing such useful feedback, helping us identify improvements for our work, especially in terms of clarity of our evaluation.
>
> > TAC is an empirical method
>
> We agree with the reviewer in that our contribution lies on the empirical side, but we remark that our motivation is both empirical and theoretical. On the empirical side, we leverage strong evidence that a simple class-dependent structure “emerges” when one trains a standard classifier. On the theory side, we claim that constraining the inner inputs of a model to lie in a small set reduces the number of valid adversarial perturbations, and rules out erroneous predictions that will not match valid internal patterns.
>
> In more detail about our theoretical contributions, the set of all adversaries is bigger than the set of the ones that match valid patterns, and the set of all erroneous predictions is bigger than the set of erroneous predictions that closely match a valid pattern, so TAC can only improve performance relative to a standard classifier, as verified empirically. The “Motivation” part of the introduction highlights the same.
>
> > experiments are suggestive, but are not extensive enough to determine whether TAC provides robust and superior decision support across a wide range of tasks, and compared to other commonly used methods like temperature calibration.
>
> We respectfully disagree with this particular comment. We highlight that most work on this kind of approach focuses on very specific tasks while, in this work, we covered several cases (e.g., [1] focuses on unseen classes, [2] on detecting misclassification errors, and [3] discriminates different datasets).
>
> Detection of unseen classes, rejecting/selective classifiers, and small norm adversarial perturbations are all considered in our empirical assessment. Also, the evaluations we report comprise both image and text classification tasks, and a wide range of model architectures is used to ensure we covered enough cases used in practical applications. We also remark that we selected recent and strong baselines.
>
> Regarding specifically temperature calibration, please note that it is orthogonal to our contribution and to the baselines we considered, and it could be used in all cases to improve calibration, though it will have no impact on rejection and prediction performance which are our focus.
>
> [1] Granese F, Romanelli M, Gorla D, Palamidessi C, Piantanida P. DOCTOR: A Simple Method for Detecting Misclassification Errors. Advances in Neural Information Processing Systems, 2021.
>
> [2] Vaze S, Han K, Vedaldi A, Zisserman A. Open-set recognition: A good closed-set classifier is all you need. arXiv preprint arXiv:2110.06207, 2021.
>
> [3] Haroush M, Frostig T, Heller R, Soudry D. A Statistical Framework for Efficient Out of Distribution Detection in Deep Neural Networks. International Conference on Learning Representations, 2021.
>
> > the value-of-classification curve (VOC) was cool
>
> We completely agree with the reviewer and we are happy to have found that piece of literature related to VOC. As models get better in terms of prediction accuracy, the community will likely focus on more subtle properties such as rejection capabilities and will then require analysis tools. We hope our paper will help popularize this framework.
>
> > metrics like baseline model accuracy and TAC model accuracy are reported for the intent classification tasks, but not for the vision tasks (e.g., CIFAR-10).
>
> Thank you for pointing this out. We added the results in the following table and included it in the appendix for completeness:
>
> |          | **Base classifier** |  **TAC** |
> |----------|:---------------:|:----:|
> |   **MNIST**  |       99.0      | 99.3 |
> | **CIFAR-10** |       95.6      | 95.0 |
> | **ImageNet** |       80.0      | 80.6 |
>
>
> > the "Det. AUCROC" metric does not seem to be described,
>
> We have added a definition of this metric in the evaluation section, following the “Area under the curve” metric description here [4]. In our case, we turn our problem into binary classification where the task is to predict whether a data instance has been classified correctly.
>
>
> [4] https://en.wikipedia.org/wiki/Receiver_operating_characteristic#Area_under_the_curve

---

> > ### Author Response · Authors · 2022-11-10
> > **Author's responses 2/2**
> >
> > > the rationale for reporting the detection rate at the point where the "true positive rate (TPR) equals the false negative rate (FNR)". Why not use the point at which the TPR and FPR are equal?
> >
> > We chose the threshold in the paper as it was demonstrated as a valid evaluation choice in related literature (cf. Equal Error Rate (EER) in [5]). We clarify that we computed the detection rates at the threshold where **false negative**  and  **false positive** (instead of **true positive** which is a typo that we fixed in the captions of Tables 2 and 3). We specifically selected this approach to compare different methods under similar conditions and avoid benefiting any particular scoring strategy.
> >
> > Please see below two Python code snippets to compute EER and the detection rate we reported to clarify that our choice matches the threshold used for EER.
> >
> > ```python
> > import numpy as np
> > from sklearn import metrics
> >
> > # This is the metric we report.
> > def compute_detection_rate(y, y_score):
> >     fpr, tpr, _ = metrics.roc_curve(y, y_score, pos_label=1)
> >     fnr = 1 - tpr
> >     t = np.nanargmin(np.abs(fnr - fpr))
> >     return tpr[t]
> >
> > # This is a standard metric in the literature.
> > def compute_eer(y, y_score):
> >     fpr, tpr, _ = metrics.roc_curve(y, y_score, pos_label=1)
> >     fnr = 1 - tpr
> >     t = np.nanargmin(np.abs(fnr-fpr))
> >     eer_low, eer_high = min(fnr[t],fpr[t]), max(fnr[t],fpr[t])
> >     eer = (eer_low+eer_high)*0.5
> >
> >     return eer
> > ```
> >
> > [5] https://www.sciencedirect.com/topics/computer-science/equal-error-rate
> >
> > > Why are all the cells for the baseline methods left blank? And what does the multicolumn column header "Individual attacks from from Auto-Attack (%)" mean?
> >
> > The online leaderboard only reports the consolidated results across attacks and it does not report the results for the individual attacks. We clarify that auto-attack comprises a set of attack models evaluated individually. However, for completeness for our models, we reported both the overall performance as well as results specific to each attack strategy. This is now mentioned in the text.

---

> > > ### Author Response · Authors · 2022-11-23
> > > **Gentle ping**
> > >
> > > Thank you again for your time and feedback. Please let us know if there's any further question you might have and if there are any issues left unaddressed. We will be happy to continue the discussion and provide further details.

---

### Author Response · Authors · 2022-11-10
**Summary**

We thank the reviewers for their work and valuable feedback. We are glad they found our method  simple and generally applicable (Reviewers sw7u and rL3k), our results strong or convincing (Reviewers sw7u, eb98, and rL3k), and our paper clearly written (Reviewer rL3k).

Based on the comments, we list here the main changes implemented in the manuscript due to improvements suggested by the reviewers:

- Included clear description of evaluation metrics and justified approach to select threshold.
- Reported accuracy for vision datasets.
- Modified motivation in the introduction to make sure that the property of interest (simple class-dependent structure in representations) is clearly described.
- Included the discussion with missing reference pointed out by Rev. eb98, and highlighted the approaches are orthogonal and could be combined.
- Included the layer-wise evaluations.
- Currently running the ECOC baseline and will include their results before the discussion period ends.

An updated version of the manuscript will be uploaded once the ECOC baselines are concluded. We addressed all comments in further detail individually to each reviewer.

---

> ### Author Response · Authors · 2022-11-14
> **Update**
>
> We have uploaded a new version of the manuscript accounting for the recommendations of the reviewers. In addition to the main changes described in the previous post, we highlight other changes that were implemented:
>
> - We added details on the codes definition.
>
> - In the evaluation Section, we added a paragraph mentioning the choices we made specifically for ImageNet, and how the general proposal serves as a good default but could be improved upon considering specific data/tasks when possible.
>
> - We added details on the auto-attack evaluation.
>
> - To open up space for all changes, we moved the layer wise distance confidence intervals to the appendix (Fig. 4 in the submitted version; now Fig. 10), and placed it together with the new layer wise results Section. We also reduced the size of Figure 2, but included a large version in the appendix for readability.

---

### Decision · Program_Chairs · 2023-01-20

**Decision:**

Accept: poster

**Justification For Why Not Higher Score:**

While the reviewers are in favor of acceptance their level of excitement does not merit a spotlight or oral according to my current self-calibration.

**Justification For Why Not Lower Score:**

All reviewers are in favor of acceptance.

**Metareview: Summary, Strengths And Weaknesses:**

Summary: authors propose a method for calibrating classifiers so that they do not return high-confidence errors.
strength: a simple method that leads to good performance.
Weakness: Some issues with clarity of writing and motivation.

**Note From Pc:**

if the above contains the word "oral" or "spotlight" please see: "oral" presentation means -> notable-top-5% and "spotlight" means -> notable-top-25%. As stated in our emails, we are disassociating presentation type from AC recommendations